# Smart soft contact lenses for continuous 24-hour monitoring of intraocular pressure in glaucoma care

Jinyuan Zhang[1,13], Kyunghun Kim[1,13], Ho Joong Kim[2,13], Dawn Meyer[3], Woohyun Park [4], Seul Ah Lee[1], Yumin Dai[5], Bongjoong Kim[1,6], Haesoo Moon[1], Jay V. Shah[7], Keely E. Harris[8], Brett Collar[9], Kangying Liu[10], Pedro Irazoqui[9], Hyowon Lee [1,11,12], Shin Ae Park [8,14] ✉, Pete S. Kollbaum [1,3,14] ✉, Bryan W. Boudouris [2,10,12,14] ✉ & Chi Hwan Lee [1,3,4,5,11,12,14] ✉

Continuous monitoring of intraocular pressure, particularly during sleep, remains a grand challenge in glaucoma care. Here we introduce a class of smart soft contact lenses, enabling the continuous 24-hour monitoring of intraocular pressure, even during sleep. Uniquely, the smart soft contact lenses are built upon various commercial brands of soft contact lenses without altering their intrinsic properties such as lens power, biocompatibility, softness, transparency, wettability, oxygen transmissibility, and overnight wearability. We show that the smart soft contact lenses can seamlessly fit across different corneal curvatures and thicknesses in human eyes and therefore accurately measure absolute intraocular pressure under ambulatory conditions. We perform a comprehensive set of in vivo evaluations in rabbit, dog, and human eyes from normal to hypertension to confirm the superior measurement accuracy, within-subject repeatability, and user comfort of the smart soft contact lenses beyond current wearable ocular tonometers. We envision that the smart soft contact lenses will be effective in glaucoma care.

Glaucoma—often called the "silent thief of sight"—gradually steals vision without early warning signs or pain, and it remains the leading cause of irreversible blindness worldwide[1,2]. Currently, the only intervention to delay glaucoma progression is to lower the intraocular pressure (IOP) of eyes under ocular hypertension, and thereby, minimize optic nerve damage (i.e., vision loss)[3,4]. Clinical trials in lowering IOP with topical ocular hypotensive medications have been successful in reducing the risk of developing glaucoma[5,6]. However, the rhythm of IOP varies by postural changes under ambulatory conditions over both nyctohemeral (i.e., 24-h) and seasonal (i.e., monthly) periods wherein overnight IOP is typically greater than daytime IOP in a supine position[7]. In turn, ocular hypertension may occur during sleep, without the patient noticing it, even if daytime in-clinic or at-home measurements indicate normal IOP (i.e., 10–21 mmHg). Therefore, glaucoma patients and suspects can benefit from evaluating the 24-h rhythm of IOP on a daily to weekly to monthly basis at home. However, none of

[1]Weldon School of Biomedical Engineering, Purdue University, West Lafayette, IN, USA. [2]Charles D. Davidson School of Chemical Engineering, Purdue University, West Lafayette, IN, USA. [3]School of Optometry, Indiana University, Bloomington, IN, USA. [4]School of Mechanical Engineering, Purdue University, West Lafayette, IN, USA. [5]School of Materials Engineering, Purdue University, West Lafayette, IN, USA. [6]Department of Mechanical and System Design Engineering, Hongik University, Seoul 04066, Republic of Korea. [7]Elmore Family School of Electrical and Computer Engineering, Purdue University, West Lafayette, IN, USA. [8]Department of Veterinary Clinical Sciences, Purdue University, West Lafayette, IN, USA. [9]Department of Electrical and Computer Engineering, Johns Hopkins University, Baltimore, MD, USA. [10]Department of Chemistry, Purdue University, West Lafayette, IN, USA. [11]Center for Implantable Devices, Purdue University, West Lafayette, IN, USA. [12]Birck Nanotechnology Center, Purdue University, West Lafayette, IN, USA. [13]These authors contributed equally: Jinyuan Zhang, Kyunghun Kim, Ho Joong Kim. [14]These authors jointly supervised this work: Shin Ae Park, Pete S. Kollbaum, Bryan W. Boudouris, Chi Hwan Lee. ✉e-mail: park1222@purdue.edu; kollbaum@indiana.edu; boudouris@purdue.edu; lee2270@purdue.edu

the current ocular tonometers are effective in continuously monitoring IOP beyond clinic hours, particularly during sleep.

The Goldmann applanation tonometry (GAT) is regarded as the current gold standard for in-clinic IOP measurements, but the need for routine clinic visits remains a significant burden and also has become a particular challenge throughout the ongoing global pandemic[8]. Portable handheld ocular tonometers, such as the I-Care Home (I-Care, Inc.), enable the at-home monitoring of IOP, but their overnight use remains limited particularly during sleep due to the requirement of patients to stay awake during measurements[9,10]. Wearable ocular tonometers, such as the TriggerFish lens (Sensimed, Inc.), enable the continuous monitoring of IOP in both in-clinic and at-home settings, but their long-term use in human eyes remains impeded particularly during sleep due to side effects including foreign body sensation, eye pain, superficial punctate keratitis, corneal epithelial defects, and conjunctival erythema[11]. Moreover, these wearable ocular tonometers are typically equipped with an integrated circuit (IC) chip for wireless communications that is formed on a rigid and stiff semiconductor wafer (i.e., like a piece of glass), thereby posing discomfort and safety risks particularly during sleep[12].

Ongoing research endeavors enable several types of smart contact lenses by integrating functional nanomaterials (e.g., graphene and metallic nanowires) with contact lenses made of biocompatible polymers (e.g., hydrogel silicones, Parylene-C, and SU-8 resins)[13-20]. These smart contact lenses have shown initial success at the laboratory scale, but their pragmatic implementation into human eyes remains impeded due to the lack of mechanical durability (for lens handling and inadvertent eye rubbing), chemical stability (for daily lens cleaning and disinfecting), oxygen transmissibility (for long-term wear), and ergonomic design (for lens fitting in human eyes with different corneal shapes and sizes)[21,22]. Moreover, most current smart contact lenses are still necessarily equipped with the stiff IC chip for wireless communications, which would thereby inevitably pose discomfort and safety risks to human eyes particularly during sleep.

On the other hand, there exist various commercial brands of soft contact lenses using ophthalmically compatible silicone hydrogels that offer a high quality of biocompatibility (i.e., FDA-cleared), softness (i.e., 0.2–2 MPa), transparency (i.e., ≥99%), oxygen transmissibility (i.e., 10–200 Dk t-1), wettability or water content (i.e., 30–80%), ergonomic curvature (i.e., 8.3–9.0 mm in base curve radii), and overnight wearability[23]. These soft contact lenses can seamlessly fit a variety of corneal shapes and sizes in human eyes without significant safety concerns for those even with glaucoma, other ocular diseases, or post-incisional surgeries[24]. Yet, it is currently challenging to fabricate a wireless ocular tonometer on commercial soft contact lenses due to the requirement of high temperatures and corrosive chemicals in current microfabrication processes that are traditionally designed for inorganic materials. Thus, there remains a critical opportunity to extend the applicability of commercial soft contact lenses into a safe and effective sensing platform for the human eye, particularly for the continuous 24-h monitoring of IOP in glaucoma care.

In this work, we introduce a unique class of smart soft contact lenses (SSCL) where a highly soft, thin, and stretchable ocular tonometer is built upon various commercial brands of soft contact lenses for the continuous 24-h monitoring of IOP even during sleep at home. Importantly, the SSCL will retain the intrinsic lens features including (1) lens power, biocompatibility, softness, transparency, wettability, and oxygen transmissibility for user comfort and clear vision; (2) overnight wearability for continuous IOP monitoring even during sleep; (3) ergonomic curvature fitting across different corneal shapes and sizes in human eyes for high measurement accuracy and within-subject repeatability; (4) mechanical and chemical durability against daily lens cleaning, disinfecting, fitting, and handling for long-term routine use; and (5) disposability after multiple uses through low-cost batch production for wide adoption in clinical practice. Having all these features

at the same time is crucial to the success of translating the SSCL into glaucoma care, but lacks in current wearable ocular tonometers.

## Results

### Device configuration and working principle

Figure 1a shows the layered schematic view of the SSCL. The internal ocular tonometer is comprised of intrinsically soft and stretchable elastomers, including: (1) polydimethylsiloxane (PDMS) for biocompatible encapsulation; (2) polystyrene-b-poly(ethylene-ran-butadiene)-b-polystyrene (SEBS) embedded with silver (Ag) flakes (AgSEBS) for conducting traces; and (3) Silbione liquid silicone rubber (Bluestar Silicones; East Brunswick, Inc.) with the mechanical modulus (E) of ~5 kPa for a highly compressible dielectric interlayer. These elastomers were directly printed into a series resistor-inductor-capacitor (RLC) resonant circuit (Supplementary Fig. 1a) through the use of an automated nozzle injection system (Nordson EFD) equipped on a three-axis computer-controlled translation stage (Supplementary Fig. 1b) in a rapid high-throughput manner (i.e., ≥30 units per batch within 30 min of printing). The resulting ocular tonometer was then bonded onto the outer surface of the soft contact lens through the in situ polymerization of a polydopamine (PDA) adhesive, which is similar to the bonding of marine mussels in nature (Supplementary Fig. 2)[25]. The in situ polymerization of the PDA adhesive is irreversible and penetrates into the upper surface of commercial soft contact lenses to form a permanent interaction, which is crucial to ensuring the mechanical durability of the SSCL for long-term routine use. Details of the materials and production process are described in the Methods section.

The underlying working principle of the ocular tonometer is as follows. A rise in IOP causes an increase in the curvature radius of an eye (i.e., approximately 3 μm mmHg⁻¹) by which the ocular tonometer is stretched and compressed in the radial and axial (i.e., thickness) directions, respectively[11]. The geometric deformation of the ocular tonometer causes an increase in inductance and capacitance in the RLC resonant circuit, which leads to a detectable decrease in resonance frequency[26]:

$$f^{-1} = 2\Pi\sqrt{LC} \qquad (1)$$

Figure 1b presents the data acquisition scheme in which the ocular tonometer is inductively coupled to a reader coil in a proximity of less than 10 mm. The reader coil can be embedded within a typical eyeglass frame or sleep eye mask for daytime and nighttime IOP monitoring, respectively. During IOP measurements, the reader coil is wired to a portable vector network analyzer (VNA) that enables the constant acquisition of reflection coefficient (S11) spectrum. Figure 1c provides a photograph of the ocular tonometer built on a commercial soft contact lens (Air Optix Night & Day Aqua; Alcon, Inc.). The ocular tonometer was thin (i.e., 50-μm thick), narrow (i.e., 200-μm wide), and configured into a serpentine (i.e., stretchable) ring shape on the peripheral area and outside surface of the soft contact lens. The inner diameter of the ocular tonometer (i.e., 13 mm) was substantially larger than the typical pupil diameter in adults (i.e., 2–8 mm) to assure clear central and peripheral vision in all directions for wearers. Figure 1d presents the cross-sectional scanning electron microscopy (SEM) view of the ocular tonometer. The AgSEBS layers were clearly separated by the Silbione layer, all of which were entirely encapsulated by the PDMS layer. In addition, the PDMS layer exhibited gradually tapered ends at an angle of 10–15° to reduce edge irritation when contact with the inner eyelid (Supplementary Fig. 3a). The resulting SSCL was fitted well to an enucleated pig eye that provides an anatomical similarity to the human eye in size and shape (Fig. 1e)[27]. The SSCL formed a highly intimate and seamless interface along the corneal anterior surface of the pig eye by virtue of the ergonomic design of its bare soft contact lens. This aspect is crucially important to maintain the measurement accuracy and repeatability of the SSCL particularly under ambulatory

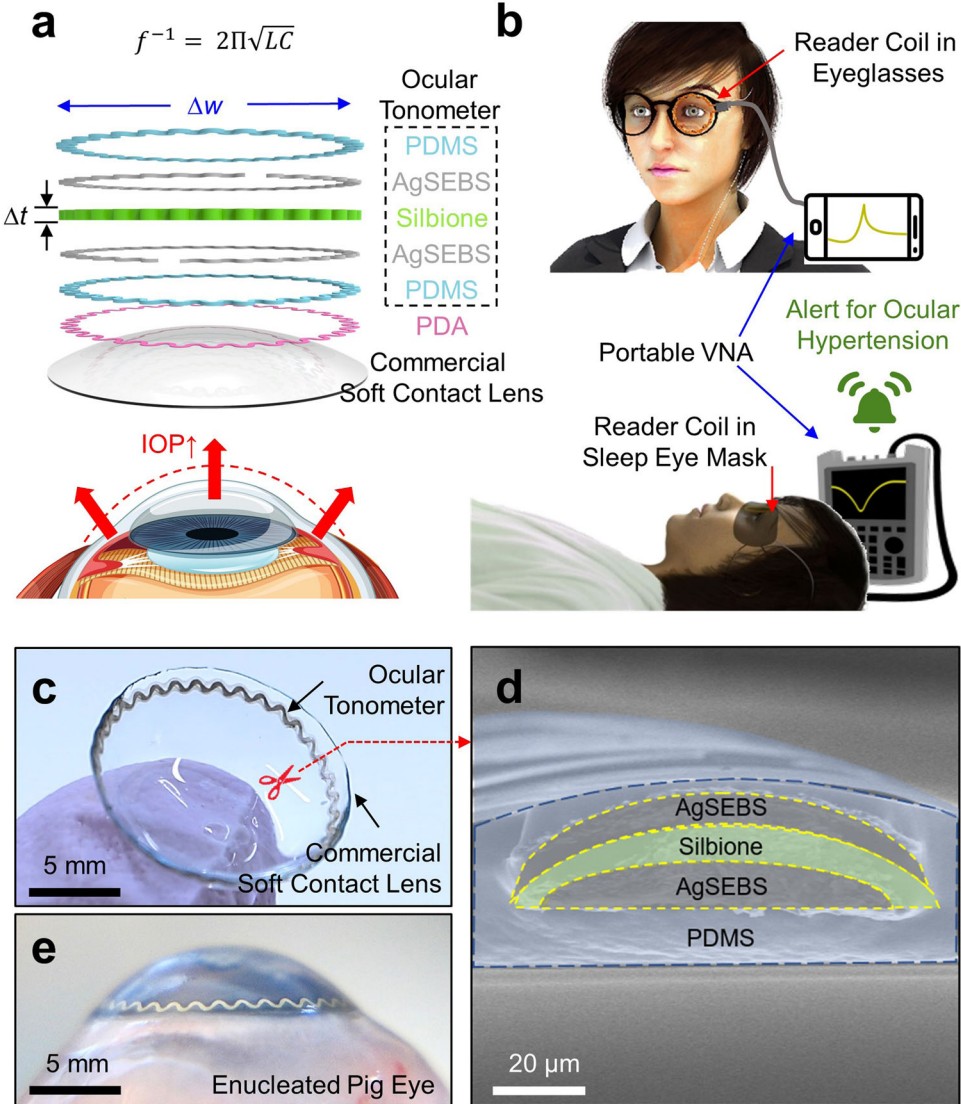

**Fig. 1 | Schematics and optical images of the SSCL. a** Layered schematic view of the SSCL. **b** Data acquisition scheme for the monitoring of daytime (top panel) and nighttime (bottom panel) IOP. **c** Photograph of the SSCL. **d** Cross-sectional SEM image of the SSCL. **e** Photograph of the SSCL in an enucleated pig eye.

conditions. Various commercial brands of soft contact lenses were applicable without a noticeable difference in overall quality (Supplementary Fig. 3b), which have been proven safe for those even with chronic ocular diseases, including glaucoma, or following incisional surgeries[24]. For all cases, the gas permeability of the SSCL remained similar to their bare soft contact lenses (Supplementary Fig. 3c).

### Benchtop evaluations

The SSCL ($E = 1.34 \pm 0.35$ MPa) was as soft as its bare soft contact lens ($E = 1.37 \pm 0.27$ MPa) mainly due to the substantial softness of the internal ocular tonometer ($E = 0.08 \pm 0.01$ MPa) (Fig. 2a). The SSCL was also stretchable in which the internal ocular tonometer remained intact without mechanical failure under stretching up to >100% even after its bare soft contact lens was torn apart at the applied strain of ~90% (Supplementary Fig. 4). Figure 2b presents that the relative resistance change ($\Delta R/R_O$) of the SSCL remained less than 10% and 30% after 10,000 cycles of stretching at 25% and 50%, respectively. Figure 2c shows the capacitance response of the SSCL under pressure in a range of 0–40 mmHg, which reflects the typical IOP range of human eyes under normal (i.e., 10–21 mmHg) and hypertension (i.e., 21–40 mmHg) conditions[28]. For comparison, different types of dielectric interlayers,

including the Silbione ($E \approx 5$ kPa), Ecoflex ($E \approx 30$ kPa), and PDMS ($E \approx 640$ kPa), were employed in the ocular tonometer. As expected, the use of the softest dielectric interlayer (i.e., Silbione) led to the highest capacitive sensitivity (i.e., $6.8 \times 10^{-4}$ mmHg$^{-1}$) over others. Details of calculating the capacitive sensitivity are described in the Methods section.

The mechanical, electrical, and chemical reliability of the SSCL is an essential consideration for its long-term use against daily lens cleaning, disinfecting, storing, handling, and misuse conditions (e.g., dehydrating and overheating). The SSCL satisfies all these needs. Specifically, the resonant frequency of the SSCL changed barely, or slightly within only ±0.2 MHz, from its baseline following multiple cycles of mechanical deformations such as flipping, rubbing, folding, and stretching (Fig. 2d); cleaning and disinfecting with commercial solutions (Renu; Bausch & Lomb, Inc. & Clear Care; Alcon Laboratories, Inc.) (Fig. 2e); 2-h dehydrating in ambient condition and 20-min rehydrating in a cleaning saline solution (Renu; Bausch & Lomb, Inc.) (Fig. 2f); 30-min significant overheating at 75 °C and 30-min cooling to −4 °C (Fig. 2g); and 30-day storing in a saline solution (Renu; Bausch & Lomb, Inc.) (Fig. 2h). A total of five measurements were taken and averaged at each data point with the error bars denoting standard

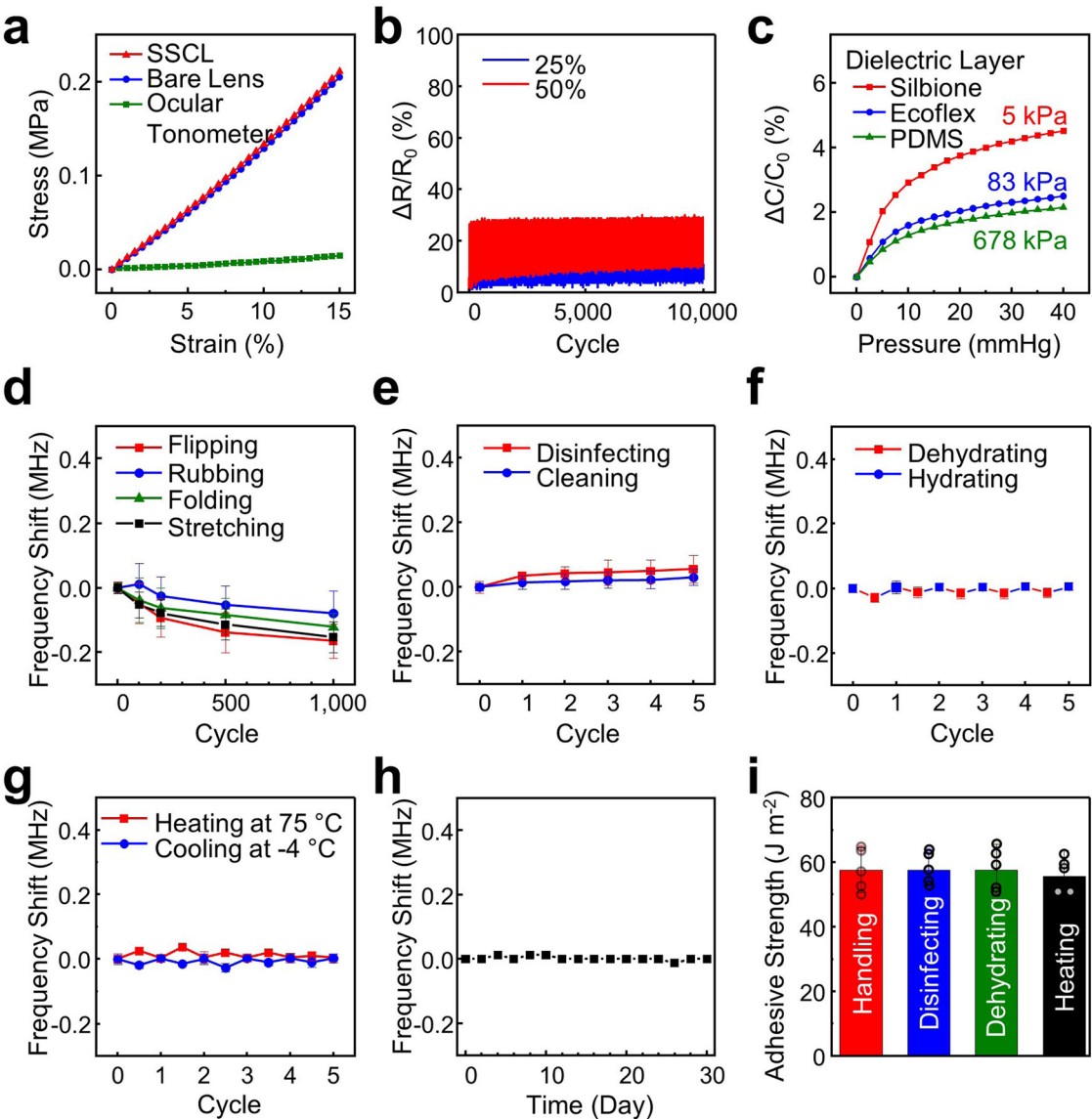

**Fig. 2 | Benchtop evaluations. a** Stress−strain curve for the SSCL (red line), the bare soft contact lens without the ocular tonometer (blue line), and the ocular tonometer without the soft contact lens (green line). **b** Resistance response ($\Delta R/R_0$) of the SSCL following 10,000 cycles of stretching at 25% (blue line) and 50% (red line). **c** Capacitance response ($\Delta C/C_0$) of the SSCL with respect to applied pressure by employing three different dielectric interlayers made of Silbione (red line), Ecoflex (blue line), and PDMS (green line) in the ocular tonometer. **d** Normalized baseline resonant frequency of the SSCL following 1000 cycles of flipping, rubbing, folding, and stretching. **e** Five cycles of disinfecting (red line) and cleaning (blue line); **f** 5 cycles of hydrating (red line) and dehydrating (blue line); **g** 5 cycles of heating and cooling; and **h** soaking in a saline solution for 30 days. **i** Adhesive strength of the PDA layer after multiple cycles of handling, disinfecting, dehydrating, and heating. The error bars represent standard deviations with $n = 5$ for each group.

deviations. Importantly, the adhesive strength of the PDA layer (i.e., $57.55 \pm 11.72\,\mathrm{J\,m^{-2}}$) was also maintained against all these conditions without any notable sign of delamination (Fig. 2i). The representative images and video of the SSCL are shown in Supplementary Fig. 5 and Supplementary Video 1, respectively.

## Ex vivo evaluations in enucleated pig eyes
The wireless sensing performance of the SSCL was evaluated ex vivo in enucleated pig eyes ($n = 3$). Figure 3a schematically illustrates the ex vivo measurement setup that includes two needles to cannulate an enucleated pig eye. The first needle was connected to a syringe pump to modulate the IOP of an enucleated pig eye by infusing and removing saline solution to/from the anterior chamber of the pig eye. The second needle was connected to a pressure gauge (V6402; Smiths Medical, Inc.) to simultaneously measure the IOP for calibration. The SSCL

was fitted to the corneal surface of the pig eye and then inductively coupled to a reader coil in a proximity of less than 10 mm (Fig. 3b). The reader coil was comprised of a wound, enamel-covered copper coil with an outer and inner diameter of 25 and 20 mm, respectively. The resistance and inductance of the reader coil were tuned at 3.5–4.0 Ω and 450–480 μH, respectively. The overall dimension and design of the reader coil were adjusted for optimal impedance matching with the SSCL. During IOP monitoring, the reader coil was wired to a portable VNA (FieldFox Handheld Analyzer 9913A; Keysight Technologies, Inc.; 292 × 188 × 72 mm). A total of six measurements were taken and averaged at each data point while the IOP of the pig eye was increased and decreased between 6 and 38 mmHg at an interval of 4 mmHg by infusing and removing saline solution, respectively.

Figure 3c presents that the reflection spectra (S11) of the SSCL were clearly shifted in response to ascending and descending IOP of

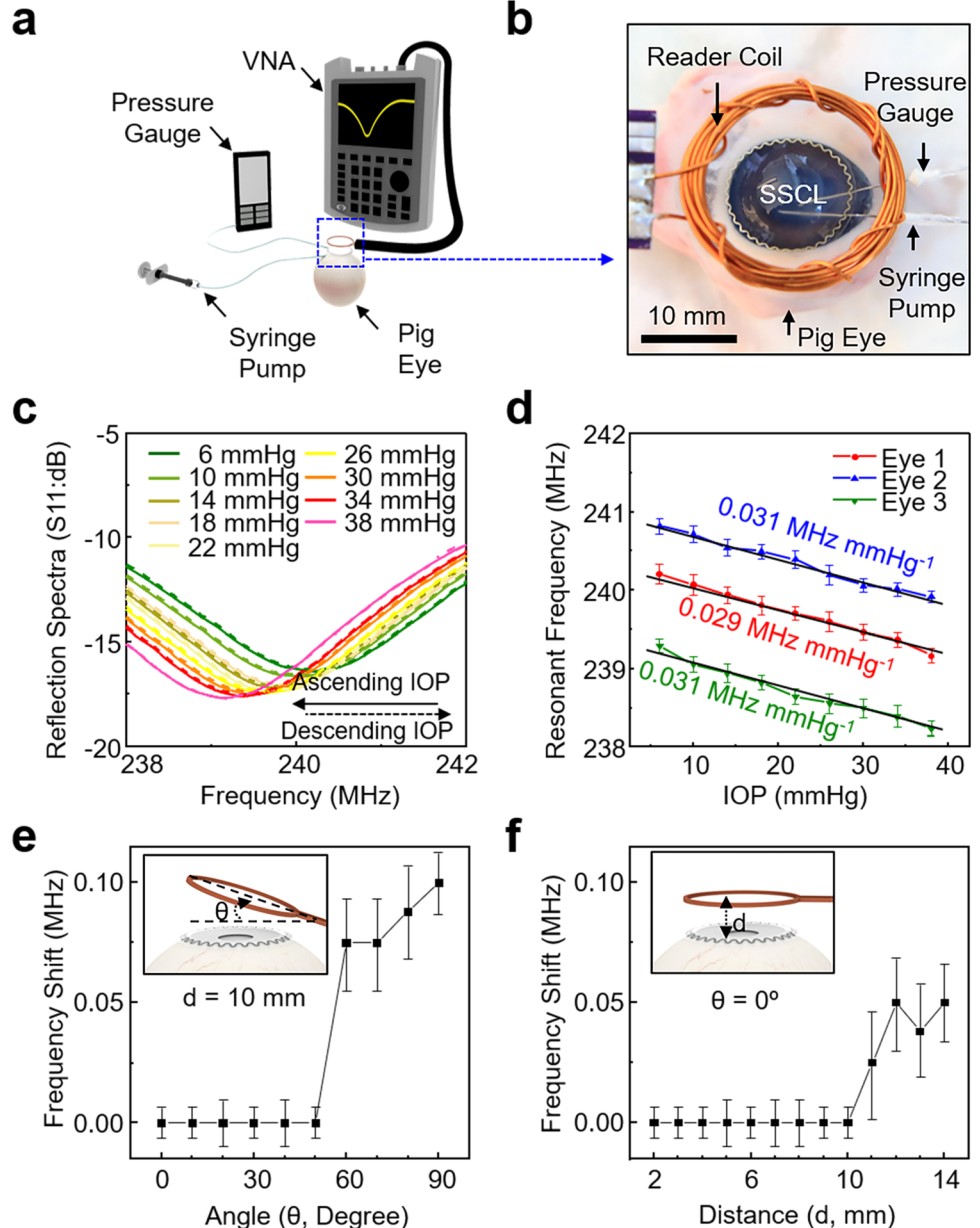

**Fig. 3 | Ex vivo evaluations in enucleated pig eyes. a** Schematic view of the ex vivo measurement setup. **b** Photograph of the SSCL in an enucleated pig eye during measurement. **c** Reflection spectra (S11) of the SSCL in response to ascending and descending IOP of the pig eye. **d** Resonant frequency of the SSCL in the response to the IOP of enucleated pig eyes ($n = 3$). **e** Baseline resonant frequency of the SSCL at various angles with respect to the reader coil ($n = 5$). **f** Baseline resonant frequency of the SSCL at various distances with respect to the reader coil ($n = 5$).

the pig eye with negligible hysteresis. Figure 3d shows the resonant frequency of the SSCL for three enucleated pig eyes with different dioptric corneal curvatures (i.e., 38.75, 38.37, and 39.37 D) and thicknesses (i.e., 948, 1084, and 1004 µm). Irrespective of the corneal curvature and thickness, an empirical linear fit was obtained for each pig eye with the corresponding responsivity and sensitivity of approximately 0.030 MHz mmHg⁻¹ ($R^2 = 0.98$) and 125 ppm mmHg⁻¹, respectively. The high measurement accuracy and repeatability of the SSCL were mainly attributed to its seamless and reliable fit to the various corneal curvatures and thicknesses of the pig eyes. For comparison, the results of control experiments with an enucleated cow eye (typical corneal curvature ≥26 D) that is dissimilar to the human eye (typical corneal curvature ≥40 D) in corneal curvature are summarized in Supplementary Fig. 6a–d. The measurement responsivity of the SSCL was considerably deteriorated due to the geometric discrepancy

(i.e., interfacial gap) between the SSCL and the cow eye. Figure 3e, f shows the effect of the angle (θ) and distance (d) between the SSCL and the reader coil on signal quality. The corresponding reflection spectra are shown in Supplementary Fig. 6e, f. No shift in the baseline resonant frequency of the SSCL appeared at θ ≤ 50° and d ≤ 10 mm, indicating that the extent of possible user misalignments or displacements can be accommodated without significant degradation in signal quality. The overall performance of the SSCL was also maintained when the reader coil was embedded within an eyeglass frame or a sleep eye mask (Supplementary Fig. 7).

## Cell viability and in vivo evaluations in rabbit eyes
The time-dependent cell viability of the SSCL to human corneal cell lines was assessed to identify any adverse response at the cellular level. Specifically, human corneal epithelial cells (HCEpiCs) were cultured in

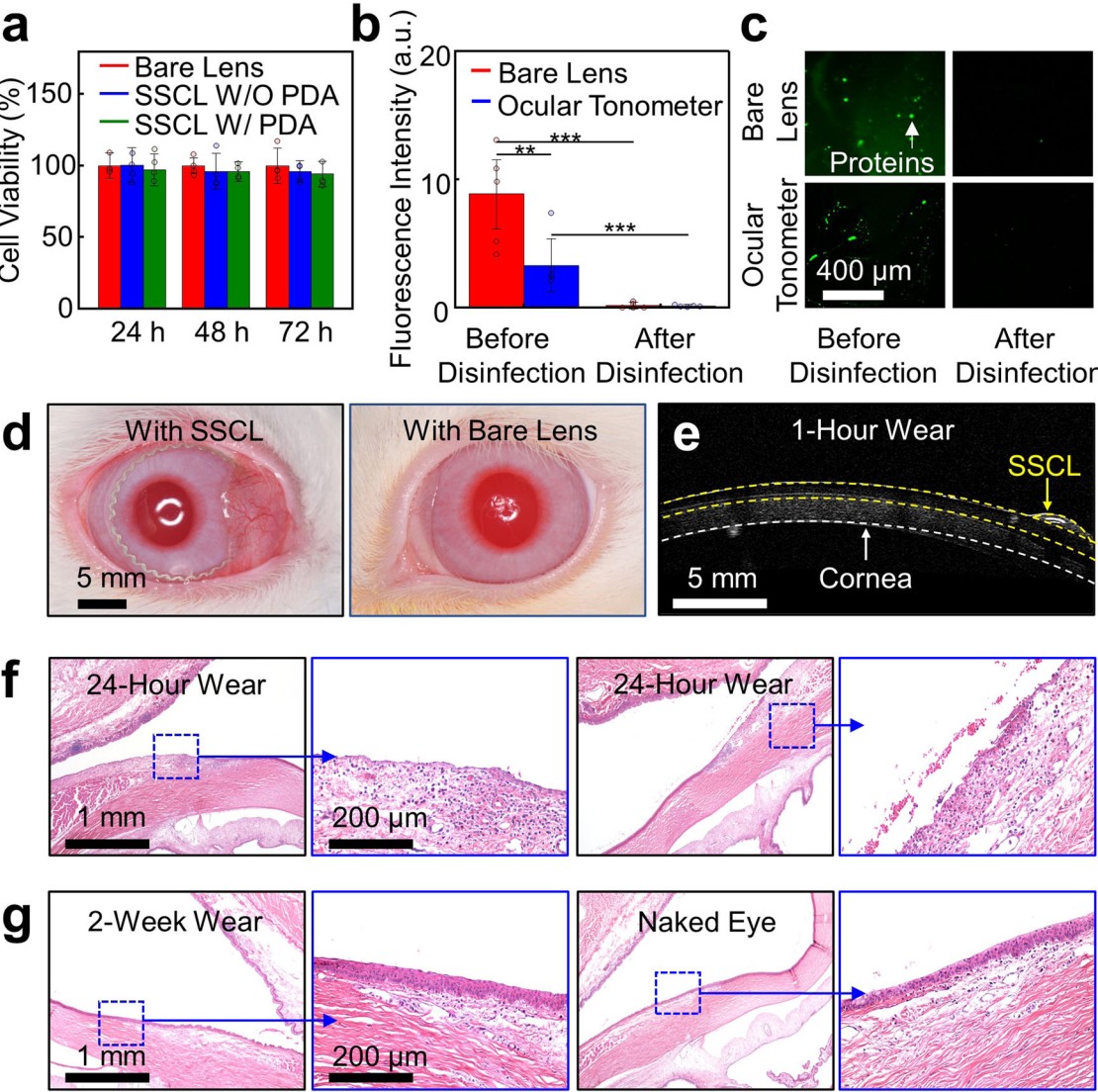

**Fig. 4 | Cell viability and in vivo evaluations in rabbit eyes. a** Cell viability assay of HCEpiCs seeded on the SSCL without (blue bars) and with (green bars) the presence of the PDA adhesive as compared to the bare soft contact lens (red bars) ($n = 4$). **b** Quantified accumulation of proteins on the bare soft contact lens (red bar) and the SSCL (blue bar) before (left column, $p = 0.02194$ before disinfecting the bare soft contact lens vs. the ocular tonometer, $n = 5$; $p = 0.00115$ before vs. after disinfecting the bare soft contact lens, $n = 5$) and after (right column, $p = 0.00979$ before vs. after disinfecting the ocular tonometer, $n = 5$) disinfection. Significance was set at \*\*\*$p < 0.01$, and \*\*$p < 0.1$. **c** Representative surface fluorescence image of the bare soft contact lens (top row) and the SSCL (bottom row) before (left column) and after (right column) disinfection. **d** Photographs of the rabbit eye wearing the SSCL (left panel) and the bare soft contact lens (right panel). **e** Representative AS-OCT image of the rabbit eye after 1 h of wearing the SSCL. **f** Representative photomicrographs of the rabbit eye with H&E staining displaying epithelial erosion and mild mixed inflammation with edema at the limbus after 24-h wear of the SSCL (left panel) and the bare soft contact lens (right panel). **g** Representative photomicrographs of the rabbit eye with H&E staining displaying no abnormality at the limbus after 2-week wear of the SSCL (left panel) as compared to those of the naked eye (right panel).

a medium (EpiGRO™ Human Ocular Epithelia Complete Media; MilliporeSigma, Inc.) at 37.5 °C for 72 h. The cells were seeded on the surface of the SSCL and measured using an MTT (3-(4,5-dimethylthiazol-2-yl)–2,5-diphenyltetrazolium bromide) assay kit (Sigma-Aldrich, Inc.). Details of the cell viability assay are described in the Methods section. Figure 4a shows the experimental results obtained from the SSCL without (blue bars) and with (green bars) the presence of the PDA adhesive as compared to its bare soft contact lens (red bars). The cell viability remained above 95% throughout the assay period without notable differences among the groups ($n = 5$ for each group). In turn, the SSCL poses little risk of developing corneal inflammation.

The biofouling resistance (i.e., protein accumulation from tear fluid) of the SSCL was also assessed, which is associated with ocular surface inflammatory complications such as giant papillary conjunctivitis[29]. Specifically, proteins were incubated on the SSCL using a 5 mg mL$^{-1}$ of bovine serum albumin-fluorescein conjugate (BSA-FITC A23015; Fisher Scientific, Inc.) in a phosphate-buffered saline (PBS; pH 7.4) for 2 h at 37.5 °C, and then, the accumulation of proteins was quantified over time via fluorescence imaging. Figure 4b presents that the accumulation of proteins over the top surface of the internal ocular tonometer of the SSCL remained lower compared to its bare soft contact lens with \*\*$p < 0.01$ according to a one-way analysis of variance (ANOVA) method with Tukey's post hoc test. The lower accumulation of proteins over the internal ocular tonometer of the SSCL is likely attributed to the presence of its hydrophobic encapsulation layer (i.e., PDMS). Importantly, the accumulated proteins were cleanly removed after a disinfecting cycle with a commercial solution (Clear Care; Alcon Laboratories, Inc.) for both groups (Fig. 4c).

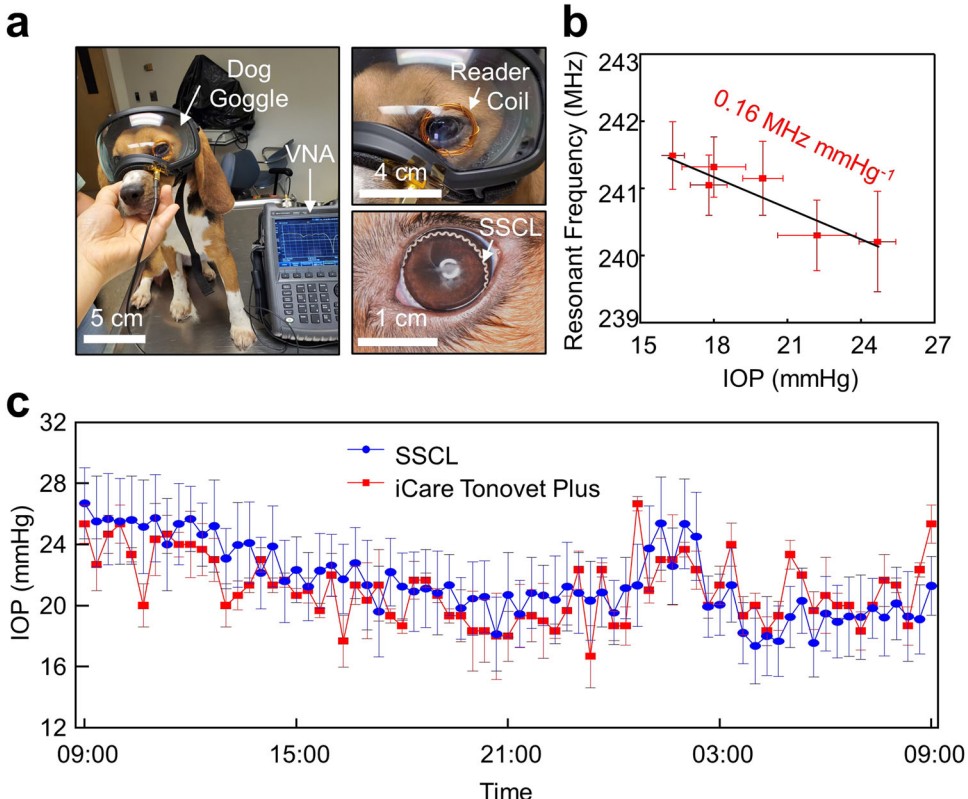

**Fig. 5 | In vivo sensing performance in dog eyes. a** Photographs of a dog wearing the SSCL and a dog goggle (V2 Goggle; Rex Specs, Inc.) embedded with the reader coil. **b** Resonant frequency of the SSCL with respect to the IOP values of the dog ($n = 3$). **c** 24-h IOP rhythm of the dog obtained from the SSCL (blue line) and the iCare Tonovet Plus (red line) ($n = 3$).

The in vivo tissue compatibility and long-term wearability of the SSCL were assessed in rabbit eyes ($n = 4$) after fitting the SSCL on an eye for 24 h as compared to its bare soft contact lens on the contralateral eye (Fig. 4d). The rabbit eye provides an anatomical similarity to the human eye in shape and size[27]. The eyelids were partially sutured (i.e., partial temporary tarsorrhaphy) to improve the lens retention at the corneal surface. Details of the experiments in rabbit eyes are described in the Methods section. Figure 4e provides a representative anterior segment ocular coherence tomography (AS-OCT) image of a rabbit eye after 1 h of wearing the SSCL, which confirmed its conformational alignment with the cornea. No notable abnormality was observed in two out of total four rabbits while mild hyperemia (grade 1; modified McDonald–Shadduck system) was noted in the palpebral conjunctiva of the other two rabbits without notable differences between the left and right eyes (Supplementary Fig. 8a). In addition, no notable abnormality was observed after routine wear of the SSCL for 8 h a day for up to 2 weeks in rabbit eyes ($n = 2$), without suturing the eyelids, as compared to the naked eyes (Supplementary Fig. 8b).

Figure 4f shows the corresponding histopathology images of rabbit eyes after 24 h of wearing the SSCL on an eye (left panel) and its bare soft contact lens on the contralateral eye (right panel). The rabbit eyes in both groups exhibited minimal to mild lesions (grade 1–2) including erosion of the conjunctival epithelium over the limbus with some edema and heterophilic or mixed inflammation. No notable differences were observed between the groups. Only minimal accumulation of heterophils and lymphocytes appeared in the palpebral conjunctiva. In addition, the rabbit eyes after the 2-week routine wear of the SSCL exhibited no or minimal lymphocytic inflammation (grade 1) in the palpebral conjunctiva without notable difference compared to the naked eyes (Fig. 4g). The minimal inflammation might be background lesions or caused by the partial suture (i.e., partial temporary

tarsorrhaphy) which is unassociated with wearing the SSCL. The quantitative assessments of histopathological inflammation grades in the rabbit cornea and conjunctiva are summarized in Supplementary Table 1. Details of evaluating the histopathology of the rabbit eyes are described in the Methods section.

## In vivo evaluations in dog eyes

The in vivo 24-h sensing performance of the SSCL was assessed in a dog eye (female beagle; 10 months old) under ambulatory conditions using a dog goggle (V2 Goggle; Rex Specs, Inc.) with a reader coil embedded (Fig. 5a). The dog eye provides an anatomical similarity to the human eye in shape and size[27]. The representative video of the dog during IOP monitoring is shown in Supplementary Video 2. For calibration, the IOP of the dog eye was also measured with a commercial ocular tonometer (iCare Tonovet Plus; iCare, Inc.) at an interval of 2 h by applying a drop of an IOP-lowering medication (0.005% Latanoprost; Bausch & Lomb, Inc.) following the first measurement. Figure 5b shows the average resonant frequency of the SSCL with respect to the IOP of the dog from six measurements at each time interval. An empirical linear fit (i.e., resonant frequency = $-0.16 \times$ IOP + 244.11) was obtained with the corresponding responsivity and sensitivity of 0.16 MHz mmHg$^{-1}$ ($R^2 = 0.88$) and 662 ppm mmHg$^{-1}$, respectively. Notably, the responsivity and sensitivity of the SSCL in the dog eye is higher than those in the enucleated pig eyes (Fig. 3d) mainly due to their fitting quality affected by different corneal rigidity and irregularity of the eyes or the ex vivo setting with the insertion of cannulation needles[30]. The standard measurement errors in the dog eye remained larger than those in the enucleated pig eye due to the continued blinking and eye movements of the dog under ambulatory conditions.

Figure 5c presents the calibrated absolute IOP values of the dog eye wearing the SSCL for 24 h (from 9 am to 9 am on the next day)

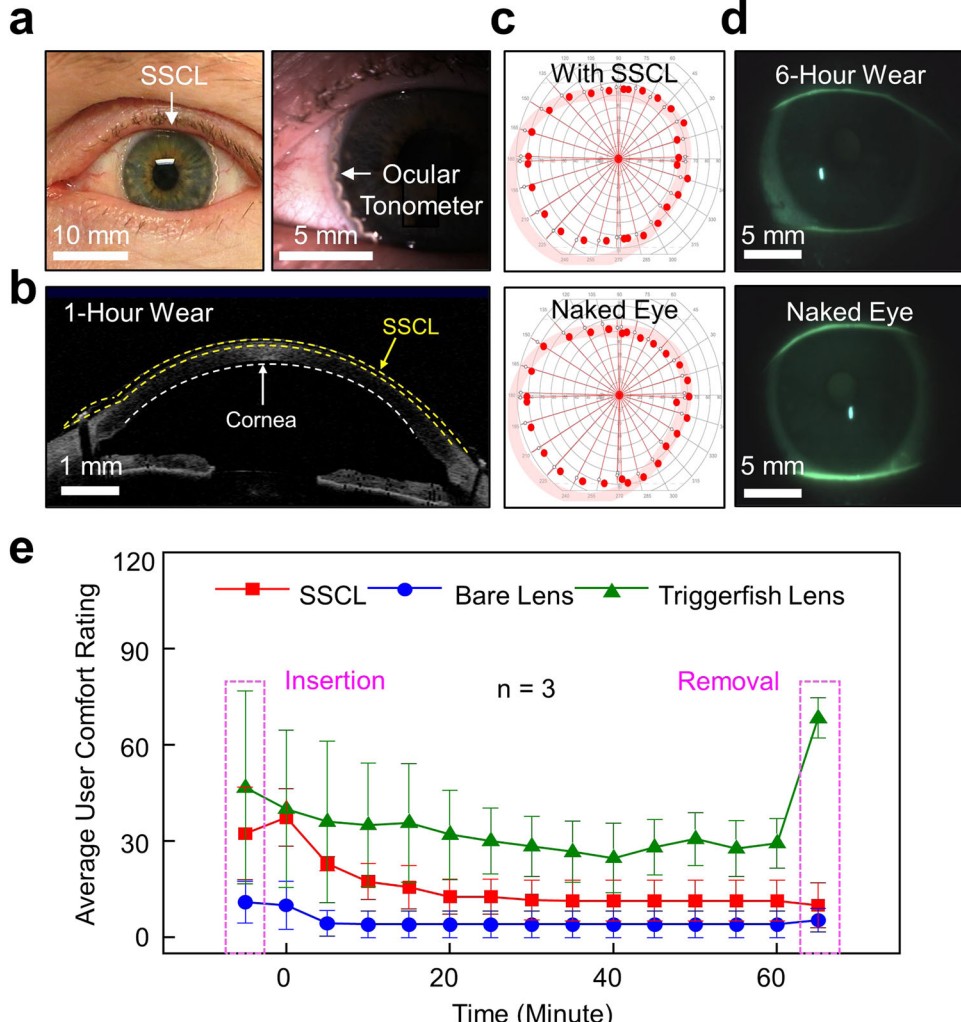

**Fig. 6 | In-clinic examinations in human eyes. a** Photograph (left panel) and slit lamp biomicroscopic image (right panel) of a human eye wearing the SSCL. **b** Representative AS-OCT image of the eye after 1 h of wearing the SSCL. **c** Representative visual field of the eye wearing the SSCL (top panel) and the naked eye (bottom panel). **d** Representative slit lamp fluorescent images of the eye after 6 h of wearing the SSCL (top panel) and the naked eye (bottom panel). **e** Average user comfort rating for the SSCL (red line), the bare soft contact lens (blue line), and the Triggerfish lens (green line) from three participants.

compared to control measurements using the iCare Tonovet Plus on the same eye in an interval of 20 min. The SSCL enabled the continuous 24-h monitoring of absolute IOP values in the dog's eye under ambulatory conditions while the iCare Tonovet Plus required the dog to remain still with its eyes open during the measurements. The measurements with the SSCL exhibited a typical IOP rhythm of the dog eye throughout the day in which its overall trends agreed well with those associated with the iCare Tonovet Plus. No notable degradation in signal quality with the SSCL occurred irrespective of the blinking or eye movements of the dog. The outliers of the IOP measurements (e.g. from 9 am to 12 pm) using the iCare Tonovet Plus were mainly attributed to motion artifacts by the excitement of the dog.

**In-clinic evaluations of human eyes**

The user comfort, ease of use, lens fit, and visual field of the SSCL were examined in human eyes. Three adults (>18 years old) who had worn contact lenses currently or previously were included in this study. Participants were tested following consent and study institutional review board approval. Prior to, during, and following each wear, the ocular health and lens fit of each participant were examined by visual acuity, participant-reported comfort rating, AS-OCT, and slit lamp biomicroscopic measurements. All participants were asked

to attend an initial in-clinic qualification visit to establish baseline measures and an initial fit of the SSCL. The lens fit, visual acuity, and user comfort rating of the SSCL were compared with wearing its bare soft contact lens or the Triggerfish lens on the contralateral eye.

Figure 6a shows the photographic (left panel) and slit lamp biomicroscopic (right panel) images of the SSCL in a participant. The insertion and removal of the SSCL from the eye of a participant is shown in Supplementary Video 3. The SSCL was fitted well on the eye with good centration and coverage similar to its bare soft contact lens. The internal ocular tonometer of the SSCL was positioned on the outer peripheral edge of the iris to be visually inconspicuous, and where the maximum strain occurred under any change of IOP[31]. The SSCL remained centered on the cornea, similar to its bare soft contact lens, during normal blinking and eye movements in all directions of gaze (Supplementary Video 4). The displacement of the SSCL on the eye was quantitatively comparable to that of its bare soft contact lens (Supplementary Fig. 9). Figure 6b provides a representative AS-OCT image of the eye after 1 h of wearing the SSCL, confirming its conformational alignment with the cornea. As expected, the visual field of the eye wearing the SSCL remained unchanged as compared to the naked eye due to the substantially larger inner diameter of the internal ocular tonometer than the pupil diameter (Fig. 6c).

Figure 6d shows the slit lamp biomicroscopic images of the eye after 6 h of wearing the SSCL (top panel) as compared to the naked eye (bottom panel) by applying a drop (~1 mg) of sodium fluorescein (Fluorets ophthalmic strips; Bausch & Lomb, Inc.). No signs of corneal damage were observed following the wear of the SSCL (Supplementary Video 5). Clinically insignificant superficial punctate corneal staining was noted in the corneal epithelial surface at a level that is commonly observed with all contact lens use[32]. The staining typically resolved within 1 h following removal of the SSCL. The bulbar and limbal conjunctiva of the eye exhibited only mild hyperemia (clinician grade ≤0.5) after >6 h of wearing the SSCL, which mirrored the response following the wear of its bare soft contact lens. No other complications were noted.

For the quantitative assessment of comfort level, each participant was asked to complete a survey during and following wear of the SSCL, its bare contact lens, and the Triggerfish lens using a simple 100-point numeric scale, where a rating of "100" represented extremely uncomfortable/intolerable and a rating of "1" perfectly comfortable/unnoticeable at all, in a similar manner as previously used in many clinical studies with wearing contact lenses[33]. Figure 6e shows the average user comfort rating following insertion, during wear, and following removal of the SSCL, its bare soft contact lens, and the Triggerfish lens at a time interval of 5 min for 60 min. All the participants were able to obtain an adequate lens fit with wearing the SSCL and settled into a satisfying comfort level. Specifically, the participants reported unnoticeable adaptation of the SSCL (i.e., average rating = $12.7 \pm 5.4$) within approximately 20 min of wear, which was slightly higher compared to the control bare contact lens (i.e., average rating = $4.0 \pm 4.2$). On the other hand, the participants reported severe discomfort during insertion (i.e., average rating = $46.7 \pm 30.1$) and removal (i.e., average rating = $68.3 \pm 6.2$) of the Triggerfish lens without noticeable stabilization over time (i.e., average rating = $24.7 \pm 10.9-68.3 \pm 6.2$).

### Ambulatory IOP monitoring in human eyes

The ambulatory IOP of the participants under postural changes was monitored in real time with the SSCL as compared to control measurements with both the iCare Home and the Triggerfish lens at time-matched points. Figure 7a, b shows representative photographs of a participant in sitting and supine postures during IOP monitoring with the reader coil embedded within an eyeglass frame and a sleep eye mask, respectively. The SSCL was able to capture the dynamic change of the IOP in response to various body postures including sitting, standing, supine, right lateral, left lateral, and prone (Fig. 7c). For calibration, the IOP of the same eye was also monitored with the iCare Home in a sequential manner immediately before each measurement with the SSCL. Six measurements were acquired in each body posture. The calibration took place only once at the initial use without iterative calibrations. The entire measurement sequence using both the SSCL and the iCare Home was completed within 1 h for each participant to minimize unwanted nyctohemeral variation in IOP. Figure 7d shows the resonant frequency of the SSCL in each body posture for the participants with different corneal curvatures (i.e., 46.2, 45.4, and 40.9 D) and thicknesses (i.e., 542, 588, and 595 μm). Participants 1 and 2 exhibited normal IOP values (i.e., 13–21 mmHg) while participant 3 exhibited mild ocular hypertension (i.e., 22–26 mmHg), which were also confirmed by the GAT. An empirical linear fit (i.e., IOP = −3.6 × resonant frequency + 890) was consistently obtained by virtue of the seamless and reliable fit of the SSCL irrespective of corneal curvature and thickness. The corresponding responsivity and sensitivity of the SSCL were 0.27 MHz mmHg$^{-1}$ ($R^2 = 0.91$) and 1121 ppm mmHg$^{-1}$, respectively. It was noted that the sensitivity of the SSCL remained substantially (i.e., by 9-fold) higher relative to those in the enucleated pig eye (Fig. 3d) mainly due to the improved fit of the SSCL to the human eye. Notably, the sensitivity of the SSCL also remained at least

2-fold higher than that (<500 ppm mmHg$^{-1}$) of current state-of-the-art wearable ocular tonometers that were tailored for human eyes (Supplementary Table 2)[34–38].

Figure 7e presents the calibrated absolute IOP values of a participant wearing the SSCL as compared to control measurements using the iCare Home and the Triggerfish lens on the same eye. The measurement results of the SSCL agreed well with those of the iCare Home, which also agreed with prior reports (Supplementary Fig. 10a)[39–42]. The IOP values of the SSCL were strongly correlated with those of the iCare Home irrespective of whether the eyelid was open or closed (Supplementary Fig. 10b). These results were reproduced in different batches of the SSCL using the same brand (Air Optix Night & Day Aqua; Alcon, Inc.) of its bare soft contact lens (Supplementary Fig. 10c). On the other hand, the measurement results of the Triggerfish lens barely agreed due to its challenge in fitting a variety of corneal curvatures and thicknesses in human eyes. The weak correlation of the Triggerfish lens (i.e., relative IOP change in mV eq) with actual IOP values in human eyes has also been observed in prior reports[11,42].

Figure 7f shows the time-varying change of the absolute IOP value in a participant wearing the SSCL for 6 h under ambulatory conditions as compared to control measurements using the iCare Home and the GAT on the same eye. The measurement results of the SSCL were closer to those of the GAT over the iCare Home. The high measurement accuracy and within-subject repeatability of the SSCL were mainly attributed to its seamless and reliable fit across different corneal curvatures and thicknesses of the eyes. Importantly, the SSCL induced no heat unlike the Triggerfish lens (Supplementary Fig. 11), and thereby can eliminate safety risks associated with ocular burn and dehydration. To avoid excessive heating of the Triggerfish lens, a sufficient time interval (i.e., 5 min) was applied between each measurement[43].

## Discussion

We report a safe and effective sensing platform for the continuous monitoring of glaucoma even during sleep at home with superior safety, user comfort, lens fit, visual field, ease of use, overnight wearability, and signal quality beyond current wearable ocular tonometers. Uniquely, this platform is built upon various commercial brands of soft contact lenses that have been proven safe for those whether or not the presence of glaucoma or following incisional surgeries. We envision that this sensing platform will be of great significance for the management of glaucoma by enabling: (1) the continuous monitoring of glaucoma progression or treatment response on a daily to weekly to monthly basis at home; (2) the immediate alert for any event of ocular hypertension even during sleep; and (3) the development of optimal treatment regimes. As potential future directions, this sensing platform can be tailored for other chronic ocular diseases such as cataract and age-related macular degeneration.

## Methods

All the research in this paper complies with all relevant ethical regulations. Animal procedures adhered to the Association for Research in Vision and Ophthalmology statement for the Use of Animals in Ophthalmic and Vision Research, and were approved by the Purdue Animal Care and Use Committee (protocol number: 2104002136). All human studies were conducted with the presence of clinical research-trained personnel in accordance with Good Clinical Practice (GCP) standards, university regulations, and approved by Institutional Review Board (IRB protocol #: 2004308902) at Indiana University.

### Device fabrication

A glass slide (Dow Corning) was cleaned by sonication in a bath of acetone and isopropyl alcohol for 30 min each and then exposed to ultraviolet ozone for 10 min. A water-soluble polyvinyl alcohol (PVA) solution (10 wt% of Mowiol 4-88 in deionized water) was spin-cast on

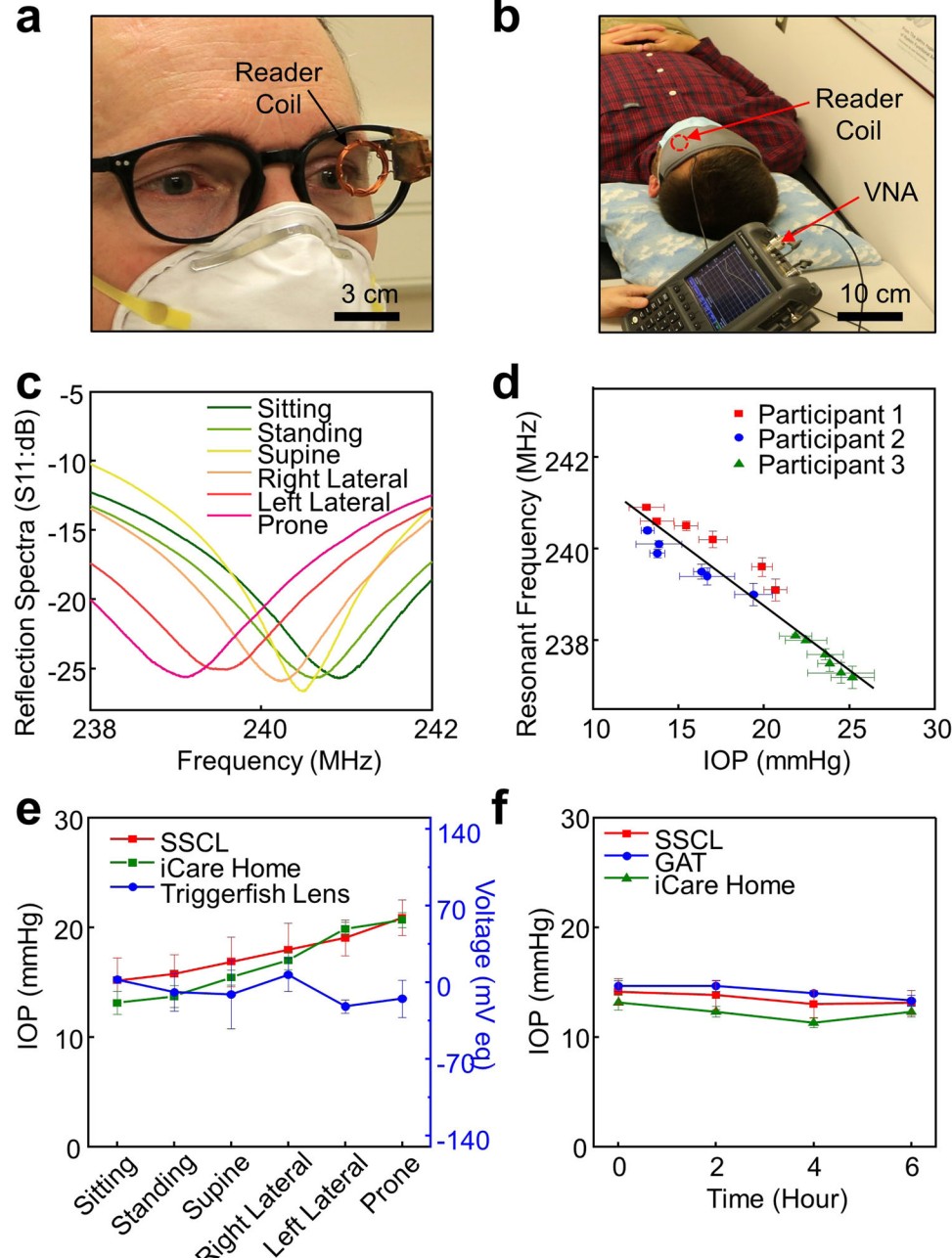

**Fig. 7 | Ambulatory IOP monitoring in human eyes. a** Photograph of a participant wearing the SSCL in a sitting posture with the reader coil embedded within a typical eyeglass frame and **b** a sleep eye mask. **c** Reflection spectra (S11) of the SSCL in response to various body postures. **d** Resonant frequency of the SSCL in each body posture for three participants with different corneal curvatures and thicknesses ($n = 3$). **e** Calibrated absolute IOP data of a participant using the SSCL (red line), the iCare Home (green line), and the Triggerfish lens (blue line). **f** Time-varying change in the ambulatory IOP of a participant obtained using the SSCL (red line), the GAT (blue line), and the iCare Home (green line) for 6 h ($n = 3$).

the glass slide at 1000 rpm for 30 s and annealed at 100 °C for 30 min. In parallel, the printable PDMS ink was prepared by mixing base solutions (Dowsil SE 1700 & Sylgard 184, Dow Corning, Inc.) and a curing agent with a weight ratio of 5:5:1. The printable AgSEBS ink was prepared by (1) dissolving SEBS (1.5 g; H1221, Asahi Kasei, Inc.) in tetrahydrofuran (1.5 g; Sigma-Aldrich, Inc.) and 1,2-dichlorobenzene (4 g; Sigma-Aldrich, Inc.) and then (2) mixing with Ag flakes (8 g; average particle size of 2–5 μm; Inframat Advanced Materials, Inc.) using a planetary centrifugal mixer (Thinky, ARE-310). The printable Silbione ink was prepared by mixing A and B components of Silbione® RT Gel4717 (Bluestar Silicones). The as-prepared inks were directly printed into a series RLC resonant circuit (i.e., ocular tonometer) on the

glass slide with a water-soluble PVA layer using an automated nozzle injection system (Nordson EFD) equipped on a three-axis computer-controlled translation stage (resolution ≥100 μm in line width; repeatability = ±3 μm; nozzle speed ≤4 mm s⁻¹)⁴⁴. The resulting structure was then immersed in a bath of deionized water to dissolve the PVA layer, allowing the ocular tonometer to be released from the glass slide. The ocular tonometer was placed afloat on the surface of an aqueous reservoir (pH = 8.5) containing 2 mg of dopamine-hydrochloride per milliliter of 10 mM Tris buffer. After approximately 12 h, thermal annealing was applied at 60 °C for 30 min to complete the in situ polymerization of the dopamine-hydrochloride into a thin layer of PDA. Here, the serpentine shape of the ocular

tonometer was stretched when interfaced with the curved surface of the soft contact lens, which thereby formed an intimate contact at the interface[45–48]. The resulting SSCL was immersed in a saline solution for 30 min and also washed with a preservative-free saline solution (Renu; Bausch & Lomb, Inc.). Prior to each test, the SSCL was disinfected with a disinfecting care solution (Clear Care; Alcon Laboratories, Inc.) for at least 6 h. The surface topology and thickness of the SSCL were characterized using an optical microscope (Eclippse LV100; Nikon, Inc.) and SEM (S-4800; Hitachi, Inc.), respectively. For comparisons, several different types of commercial soft contact lenses were tested in this study, including (1) Air Optix Night & Day Aqua (Alcon, Inc.); (2) Acuvue Oasys (Johnson & Johnson, Inc.); (3) Biofinity (CooperVision, Inc.); and (4) Dailies Total 1 (Alcon, Inc.).

## Benchtop evaluations

Standard tensile tests were conducted to determine the elastic moduli of the SSCL, its bare soft contact lens, and the internal ocular tonometer (without a soft contact lens). The specimens were loaded on the chuck of a tensile testing machine (ESM303, Mark-10) and then stretched from 0 to 15% at the elongation rate of 10–25% per minute. The stress was computed by the equation:

$$\sigma = FD^{-1}T^{-1} \qquad (2)$$

where $F$ is the measured force during the elongation; and $D$ and $T$ are the diameter and thickness of the specimen, respectively. Five measurements were taken and averaged for each sample. For the measurement of the capacitive response, the SSCL was connected to an LCR meter (E4980AL; Keysight, Inc.) via a conducting (i.e., AgSEBS) wire while increasing the pressure by applying weights (3.3 g or 2.5 mmHg). The capacitive sensitivity ($S$) of the SSCL is defined by:

$$S = \frac{\delta((C_2 - C_1)/C_0)}{\delta(P_2 - P_1)} \qquad (3)$$

where $C_2$ and $C_1$ are the capacitance acquired at sampling events 2 and 1, respectively; $C_0$ is the initial capacitance; and $P_2$ and $P_1$ are the pressure acquired at sampling events 2 and 1, respectively. For the measurement of the relative resistance change, the SSCL was connected to a source meter (Keithley 2400; Keysight, Inc.) via a conducting (i.e., AgSEBS) wire during cyclic stretching up to 10,000 times at the applied strain of 25% and 50% with the elongation rate of 5% per second, respectively. The relative resistance change of each specimen was defined by:

$$R_{\text{Rel}} = \Delta R/R_0 \qquad (4)$$

where $\Delta R$ is the difference between current resistance and initial resistance; and $R_0$ is the initial resistance value.

## Gas permeability tests

The gas permeability of the SSCL was assessed by measuring water vapor transmission[49]. Vials filled with 1 mL of deionized water were covered with the dehydrated SSCL and the dehydrated bare soft contact lenses of various commercial brands. The gap between the vials and specimens was sealed with Kapton® tape. For comparison, vials filled with the same amount of deionized water were also left uncovered and fully covered with a VHB™ tape (3M, Inc.). The weight loss of the water in each vial was measured for 186 h with an interval of approximately 24 h. Three measurements were taken and averaged for each specimen.

## Adhesive strength measurements

The adhesive strength of the PDA adhesive from the bare soft contact lens was measured using a custom-modified mechanical peeling apparatus (ESM303; Mark-10; high-resolution force gauge of ±0.25%). The bare soft contact lens was cut into the dimension of 4 × 10 mm and bonded onto the PDMS substrate (i.e., the encapsulation layer of the SSCL) using the PDA adhesive. The PDMS substrate was fixed on a horizontal stage of the mechanical peeling apparatus with a double-sided Kapton tape. The specimen was then pulled vertically at a speed of 0.4 mm s⁻¹. During the peeling test, both the peeling displacement and force were recorded. Seven measurements were taken and averaged for each specimen. The peeling strength was defined by:

$$W = \frac{F}{L} \qquad (5)$$

where $F$ is the applied force; and $L$ is the contact length. The measurements of the specimens were repeated after (1) disinfecting with a cleansing kit containing a 3% hydrogen peroxide ($H_2O_2$) cleansing solution (Clear Care; Alcon, Inc.) for 12 h; (2) dehydrating in an ambient environment (relative humidity = 40%; temperature = 25 °C) for 5 h; and (3) overheating in a saline solution maintained at 75 °C for 30 min.

## Extreme use condition tests

The change in the baseline resonant frequency of the SSCL was monitored against various user mishandling conditions. Three specimens were tested for five cycles of each condition. Six measurements were taken and averaged for each iteration. For the mechanical deformation tests, the specimens were repeatedly flipped, rubbed, folded, and stretched using an automated tensile testing machine (ESM303, Mark-10) for up to 1000 times each. In the flipping tests, the specimens were fixed under the tensile testing machine by two grips. Upper and lower probes were driven downwards and upwards by the tensile test machine at a pre-programmed speed of 180 mm min⁻¹. In the rubbing tests, the specimens were conformally fitted on a plastic artificial eyeball (Tanlee) which was fixed on the lower grip of the tensile testing machine. An artificial finger covered with a latex glove was connected to the upper grip and positioned to be in contact with the marginal region of the specimens. The artificial finger was driven to repeatedly rub the surface of the internal ocular tonometer at a speed of 120 mm min⁻¹. In the folding tests, the lower and upper margins of the specimens were fixed by the lower and upper grip of the tensile testing machine, respectively. The specimens were repeatedly folded at the applied strain of 50% through the cyclic linear motion of the upper grip at a speed of 120 mm min⁻¹. In the stretching tests, the specimens were repeatedly stretched at the applied strain of 15% through the cyclic linear motion of the upper grip at a speed of 120 mm min⁻¹. During the measurements, the specimens were constantly hydrated by applying a saline solution in every 10 min. For the disinfecting and cleaning tests, the specimens were repeatedly disinfected and cleaned up to five times with 3% $H_2O_2$ cleansing solution (Clear Care; Alcon, Inc.) and cleaning saline solution (Renu; Bausch & Lomb, Inc.), respectively. For the dehydrating tests, the specimens were placed in an ambient condition (relative humidity = 40%; temperature = 25 °C) for 1 h and then rehydrated in a saline solution for 20 min. For the overheating tests, the specimens were repeatedly heated in an oven at 75 °C for 30 min and then rested at room temperature for 1 h. For the overcooling tests, the specimens were placed in a refrigerator at −4 °C for 30 min and then rested at room temperature for 1 h.

## Ex vivo evaluations in enucleated pig eyes

The enucleated pig eyes ($n = 3$) were horizontally placed on the table. The central corneal thickness and the corneal curvature of the pig eyes were measured using a pachymeter (Pachmate 2; DGH) and Keratometer, respectively. The anterior chamber of the pig eyes was cannulated with a needle for injecting or removing a saline solution to increase or decrease IOP. A pressure transducer (V6402; Smiths Medical, Inc.) was used to monitor the IOP of the eyes as a reference.

An infusion needle was connected to a microfluidic syringe pump system (Ultra Micro Pump 3; WPI) to dynamically control the IOP of the pig eyes by the amount of the injected saline. The pressure transducer was connected to a data acquisition platform (USB-6343; National Instruments, Inc.) to instantaneously transport the data to an external computing unit. The real-time IOP variation was plotted using a MATLAB script. The SSCL was conformally fitted on the cornea of the pig eye with a reader coil fixed above the SSCL at a proximity of <10 mm. The reader coil was connected to a VNA (FieldFox Handheld Analyzer 9913A; Keysight Technologies, Inc.) to constantly acquire the frequency spectra of the reflection (S11) from the SSCL. Prior to each measurement, the IOP of the pig eyes was stabilized at 6 mmHg for 1 min. The IOP was increased by the microfluidic syringe pump at an interval of 4 mmHg until the IOP reached 38 mmHg. After stabilizing at 38 mmHg for 1 min, the IOP was decreased by the microfluidic syringe pump at an interval of 4 mmHg until the IOP reached 6 mmHg. A total of six measurements were taken and averaged at each step. The responsivity of the SSCL was defined by:

$$R = |\frac{\partial f}{\partial P}|_{P = P_{\min}} \qquad (6)$$

where $f$ is the resonant frequency; $P$ is the IOP value; and $P_{\min}$ is the minimum IOP value. The sensitivity of the SSCL was defined by:

$$S = \frac{R}{f(P = P_{\min})} \qquad (7)$$

where $R$ is the responsivity; and $f(P = P_{\min})$ is the resonant frequency at the measured minimum IOP value.

## Cell viability tests

The specimens ($n = 5$ each) were sterilized with a mixture of ethanol and distilled water (70:30 v/v) for 30 min; rinsed with Dulbecco's PBS (Gibco); and dehydrated with ultraviolet irradiation for 1 h. The specimens were then placed inside a 24-well plate on a concave side facing upwards. HCEpiCs (MilliporeSigma, Inc.) with a density of $1 \times 10^5$ per well were seeded on the specimens in a cell media (EpiGRO™ Human Ocular Epithelia Complete Media; MilliporeSigma, Inc.) and then incubated at 37.5 °C with 5% carbon dioxide ($CO_2$) for 24, 48 and 72 h. At each time point, recurring well plates from the experiment and control group were taken out from the incubator for cell viability measurement. A 3-(4,5-dimethylthiazol-2-yl)–2,5-diphenyltetrazolium bromide (MTT; MilliporeSigma, Inc.) reagent was added to each well plate and incubated for 3 h; the cell media was removed; and the cells were lysed with dimethylsulfoxide (ATTC). The light absorbance of each well was measured using a microplate reader (Synergy™ NEO; BioTek, Inc.) at a wavelength of 575 nm. The cell viability values were obtained by normalizing the absorbance values with those of the control groups. The statistical analysis was carried out using a one-way ANOVA method (****$p < 0.0001$) with Tukey's post hoc test implemented in the Origin software (OriginLab) and is expressed as averages ± s.e.m ($n = 5$).

## In vivo evaluations in rabbit eyes

All animal procedures adhered to the Association for Research in Vision and Ophthalmology statement for the Use of Animals in Ophthalmic and Vision Research, and were approved by the Purdue Animal Care and Use Committee (protocol number: 2104002136). Six female New Zealand white rabbits (6-month-old; Envigo Global Services, Inc.) were included. The rabbits were housed in an animal facility with a 12-h light/dark cycle. Four out of six rabbits were used for the 24-h biocompatibility study. The SSCL was fitted on one eye while its bare soft contact lens was fitted on the contralateral eye. Complete ophthalmic examinations including Schirmer tear test (Schirmer tear test strip;

Merck Animal Health, Inc.), tear film breakup time, fluorescein staining (Ful-Glol; Akorn, Inc.), tonometry (iCare Tonovet Plus; iCare, Inc.), slit lamp biomicroscopy (SL-17; Kowa, Inc.), and indirect ophthalmoscopy (Keeler, Inc.) were performed prior to and after the 24-h wear of the SSCL and its bare soft contact lens. Any noted abnormality was recorded using a modified Mcdonald–Shadduck semi-quantitative scoring system[50]. AS-OCT (Spectralis; Heidelberg Engineering, Inc.) images were obtained after approximately 1 h of wearing the SSCL to confirm their conformational alignment with the cornea. After the SSCL or its bare soft contact lens was fitted, the eyelids were partially sutured closed (i.e., partial temporary tarsorrhaphy) with a 4-0 nylon (Ethilon; Ethicon, Inc.). The rabbits were individually housed in each cage and allowed for normal daily activities for 24 h. The rest of the two rabbits were used for the 2-week biocompatibility study. Following the baseline complete ophthalmic examinations, the SSCL was fitted on one eye while the contralateral eye served as untreated control. The SSCL was maintained on the eyes for 8 h per day for 2 weeks. A new clean SSCL was used each day. The rabbits were individually housed in each cage and allowed for normal daily activities. The eyes were monitored daily to inspect for any sign of irritation or inflammation. Complete ophthalmic examinations were followed on days 3, 7, and 14 as compared to their baseline results. At the end of the 24-h and 2-week biocompatibility studies, the rabbits were humanly euthanized. The eyes and adnexa including upper, lower, and third eyelids were collected. The collected tissues were routinely fixed, processed in paraffin, sectioned, and stained with H&E for histopathological analysis and reporting. The histopathology was assessed in a masked fashion using a semi-quantitative scoring system ranging from 0 to 4 (no lesions, minimal, mild, moderate, severe)[51,52].

## In vivo evaluations in dog eyes

All animal procedures adhered to the Association for Research in Vision and Ophthalmology statement for the Use of Animals in Ophthalmic and Vision Research, and were approved by the Purdue Animal Care and Use Committee (protocol number: 2104002136). The SSCL was calibrated with the iCare Tonovet Plus in a dog eye (female beagle; 10 months old) to generate an empirical equation with a linear curve fitting. For calibration, the IOP of the dog was monitored with the iCare Tonovet Plus followed by the SSCL in an interval of 2 h after applying a drop of IOP-lowering medication (0.005% Latanoprost; Bausch & Lomb, Inc.) following the first measurement. A total of six measurements were taken and averaged at each time interval to obtain an empirical linear fit. On a different day, the IOP of the dog was continuously monitored for 24 h using the SSCL and the iCare Tonovet Plus in an interval of 20 min. The dogs were not sacrificed. The dog was easier to handle over rabbits for IOP monitoring particularly under ambulatory conditions.

## Heat generation in long-term operation

An enucleated pig eye was horizontally placed with the SSCL fitted on its cornea. A reader coil was placed above the SSCL in a proximity of 5 mm and also connected to a portable VNA. The pig eye was monitored using a high-resolution infrared camera (A300; FLIR) while the SSCL was under long-term operation for 12 h.

## Ocular coherence tomography (OCT)

The OCT images were acquired prior to and during the application of the SSCL or its bare soft contact lens. The data acquisition was made in corneal high-resolution mode at 16 meridians (e.g., slices).

## Slit lamp biomicroscopy

The movement and centering of the SSCL on human eyes during typical eye activities were evaluated using a slit lamp biomicroscope (SL120; Zeiss). A millimeter reticle in the ocular microscope was utilized for measurements, which was calibrated at ×10 magnification.

The eye movements in nasal, temporal, superior, and inferior directions as well as a blink and lens lag upon primary and up gaze lens were evaluated and compared with those of wearing its bare soft contact lens. Prior to and after the application of the SSCL in human eyes, the ocular health of the anterior segment eye (e.g., lids, lashes, conjunctiva, cornea) was also evaluated by a slit lamp biomicroscope examinations with and without sodium fluorescein (1 mg; Fluorets ophthalmic strips; Bausch and Lomb, Inc.). A cobalt blue light was applied. Sodium fluorescein staining was used on the eye before and after 6 h of wearing the SSCL to stain dead or devitalized cells of the cornea and conjunctiva. Magnifications ranging from ×10 to ×16 were used.

### Visual field tests
Visual field extent of human eyes was quantified for the naked eye and the same eye wearing the SSCL. The visual field extent was measured using the Octopus 900 Kinetic Perimeter (Haag-Streit) with a III4e target (i.e., area = 64 mm$^2$; luminance = 318 cd m$^{-2}$) moving at 5° per second.

### Human subject studies
All human studies were conducted in the presence of clinical research-trained personnel in accordance with GCP standards, university regulations, and approved by Institutional Review Board (IRB protocol #: 2004308902). The human research participants consented to the publication of photos. There was no participant compensation. Three adult participants (i.e., ages 46, 32, and 23) were included. The measurements with the SSCL and the iCare Home were completed within 2 h to avoid unwanted variations of IOP by nyctohemeral rhythm. At each posture, the data were collected after the IOP of each subject was stabilized by staying still and relaxed for 5 min. A total of six measurements were taken and averaged at each posture for both opened and closed eyes. Four devices were applied on participant 1; and one device was applied on participants 2 and 3 each. The continuous monitoring of the IOP for participant 1 was conducted at sitting posture by the SSCL and the GAT for 6 h in an interval of 2 h. A total of six measurements were taken and averaged for each data point. The measurements with the Triggerfish lens on each subject were conducted at the same time of day on the second day of the measurement with the SSCL to minimize the unwanted variations of IOP by nyctohemeral rhythm. At each posture, the participants stayed still and relaxed until the IOP data were collected automatically every 5 min by the Triggerfish lens. An empirical linear calibration equation was obtained using the SSCL from the participants by capturing the dynamic change of the IOP in response to various body postures including sitting, standing, supine, right lateral, left lateral, and prone. For calibration, the IOP values of the participants were also obtained using the iCare Home at each body posture.

### Statistics and reproducibility
Methods for statistical analyses have been described in detail in previous Methods subsections.

### Reporting summary
Further information on research design is available in the Nature Research Reporting Summary linked to this article.

## Data availability
All relevant data supporting the key findings of this study are available within the article and its Supplementary Information files or from the corresponding author upon reasonable request. Source data are provided with this paper.

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

## Acknowledgements

We thank Harm HogenEsch for histopathological analysis and providing histopathology images and descriptions. C.H.L. acknowledges the Leslie A. Geddes Endowment at Purdue University. H.J.K. and W.P. acknowledge the Leslie Bottorff Fellowship Program at Purdue University. Publication of this article was funded in part by Purdue University Libraries Open Access Publishing Fund. The animal study efforts of S.A.P. were supported by the National Institute of Health (NIH) (Award Number: K08EY030950). The polymer synthesis efforts of H.J.K. and B.W.B. were supported by the U.S. Department of Energy (DOE), Office of Science, Basic Energy Sciences (BES), under Award DE-SC0021967, and they gratefully thank DOE for this support.

## Author contributions

P.S.K., B.W.B., and C.H.L. conceived the concept, planned the project, and supervised the research. J.Z., K.K., H.J.K., S.A.L., Y.D., B.K., K.L., B.W.B., and C.H.L. designed, fabricated, characterized, and implemented the device. D.M. and P.S.K. designed and conducted human subject studies. K.E.H. and S.A.P. designed and conducted animal studies. B.C., J.V.S. and P.I. designed, fabricated, characterized, and implemented the wireless communication circuit. W.P. conducted the cell compatibility tests. H.M. and H.L. designed and conducted the biofouling studies. J.Z., K.K., H.J.K., S.A.P., P.S.K., B.W.B., and C.H.L. wrote the manuscript. All authors commented on the paper.

## Competing interests

The authors declare no competing interests.
