## [Peer Review File · Nature Communications]

REVIEWER COMMENTS

Reviewer #1 (Remarks to the Author):

This paper presents a smart soft contact lens sensor (SCLS) where a stretchable ocular tonometer is built on a commercial overnight-wearable contact lens for long-term intraocular pressure (IOP) measurements. The SCLS, which is composed of PDMS, AgSEBs, Silbione and PDA wet adhesive, is soft, stretchable, conformal, and has high measurement accuracy. The novelty of this work comes in its capability to achieve 24 hours safe monitoring of IOP, even during sleep. Compared to previous work, the SCLS used commercial contact lens as substrate, which is inherently safer. This work demonstrated the application on dog and human measurements under ambulatory situations. It can be considered for publication in Nature Communications. However, there are several minor revisions that need to be done and several technical points need to be addressed.

1. Page6, Line 75. & Figure S2.

It seemed that PDA is a good choice of wet adhesive which only needs 30 minutes heat treatment for attachment. Please give an explanation of the adhesive mechanism via heat.

2. Page8, Line117-120.

I'm wondering the meaning of the capacitive sensitivity. Please explain the meaning and calculation method of "highest capacitive sensitivity ($6.8 \times 10^{-4} \text{ mmHg}^{-1}$)".

3. Figure 2d & Page8, Line125

The label of Figure 2d is incorrectly written as 2c.

It seemed that there is an obvious shift after stretching, folding and flipping. Statistical analysis method should be used to clarify the differences. The meaning of error bars need to be defined. The claim "maintained without a substantial shift from baseline" may be not appropriate without statistical significance analysis.

4. Page 11. P191 & Figure 4b

"the accumulation of proteins on the SCLS remained lower compared to the bare soft contact lens" Why does this happen?

5. Figure 2f and Figure 2g.

Please quantify the results in figure 2f and 2g. While the image shows no significant change in eye response, a quantification of the damage is required.

6. Page11, Line205

The authors describe that after 2 weeks wear (8 hours per day), no or minimal lymphocytic inflammation occurred. And the damage might be background lesions unassociated with the application. In this test, a new clean SCLS was used each day. And the 24 hours wearing resulted grade 1 or 2 lesions. Does that mean the safe duration of SCLS is 8 hours per day? And we know the need for monitoring IOP is more than 8 hours per day.

7. Why choose the different animal model for biocompatibility test(rabbits) and IOP monitoring test(dog)? Are there any special reasons?

8. Figure 5c. Statistical analysis are supposed to be done for significance of differences between SCLS and iCare, while it seemed there is a large difference at some timepoint, such as 11:00, 9:00.

Reviewer #2 (Remarks to the Author):

Excellent paper reporting significant progress in the monitoring of IOP using a passive sensor integrated onto a soft contact lens.

The basic idea of a passive RLC sensing device that can be wirelessly interrogated by examining its resonance frequency is similar to what was presented earlier by Ophtimalia (France), but the authors have moved the technology forward so the device is compatible with soft contact lenses.

The authors did a thorough job in assessing the device's performance and the basic biocompatibility aspects, also in relation to the competing 'Triggerfish' lens (Sensimed, Switzerland).

I only have minor remarks regarding this manuscript:

Fig. 2. Benchtop Evaluations: label 'c' is used twice; label 'd' is missing.

Line 313: is the improved fit really the major reason for the higher sensitivity? Or is it the difference in rigidity of an ex vivo versus an in vivo eye as suggested on line 231?

Typo:

line 203: 'ribbits' → 'rabbits'

Methods:

(Benchtop evaluations) In the description of how the relative resistance variation (as a result of a number of stretching cycles) and the capacitive response under pressure variations were measured, it is not clear how the AgSEBS layers were electrically contacted.

Reviewer #3 (Remarks to the Author):

Dr. Chi Hwan Lee provided extensive data and results on testing their "smart" contact lens tonometer using an off-the-shelf contact lens. Overall the noteworthy results include:

- 1) The explanation of the "smart tonometer" is clearly explained with encapsulation material, conductive material, adhesives, and reader coil system.
- 2) Ex vivo studies were conducted to compare IOPs between the "smart contact lens" on whole perfused pig eyes cannulated with a sensor, which is proper method design. It is surprising that the commercial contact lens worked on the pig eyes given the oval shape of their corneas.
- 3) Pre-clinical data is provided in rabbits, beagle with correlation of IOPs measured with the Tonovet tonometer. While not mentioned, it is assumed the commercial contact lens fit reasonably well for the ex vivo studies, rabbit (i.e., with a tarsorrhaphy) and beagle. The animal models are a standard choice for methods.
- 4) Proper safety experiments were conducted in vitro and in vivo on the rabbits and in the pilot studies on humans.
- 5) Durability of the "smart contact lens tonometer" to cleaning, stretching, manipulation of bending to place and remove the contact lens, temperature tolerance, and various contact lens solutions.
- 6) Use of non-standardized survey to compare comfort of "smart lens" with standard contact lens and Trigger fish. As a reviewer, I believe that this data should be part of the results and not as supplement.
- 7) Supplementary videos are provided.
- 8) The supplementary figures are numerous; can any of them be combined?

The work presented in this manuscript has the potential to be of great significance for patients with glaucoma to assess IOP variability in the real world beyond limited clinic hours, treatment response to glaucoma medications, treatment response to glaucoma laser, and even in certain glaucoma post-operative settings.

Minor issues include:

a) Word choice of "exploded" to describe the "smart lens" composition. Alternative word choices include: "stacked", "layered", "integrated".

b) A more standard survey to assess comfort would be the use of a Likert scale.

c) For supplementary Fig 15, it is not clear what the two dashed lines are for; are these confidence intervals?

REVIEWER 1

Overall Comments: This paper presents a smart soft contact lens sensor (SSCL) where a stretchable ocular tonometer is built on a commercial overnight-wearable contact lens for long-term intraocular pressure (IOP) measurements. The SSCL, which is composed of PDMS, AgSEBs, Silbione and PDA wet adhesive, is soft, stretchable, conformal, and has high measurement accuracy. The novelty of this work comes in its capability to achieve 24 hours safe monitoring of IOP, even during sleep. Compared to previous work, the SSCL used commercial contact lens as substrate, which is inherently safer. This work demonstrated the application on dog and human measurements under ambulatory situations. It can be considered for publication in Nature Communications. However, there are several minor revisions that need to be done and several technical points need to be addressed.

Our Response: We thank the reviewer for the favorable comments and the recommendation for its publication in this journal.

Comment #1: Page 6, Line 75. & Figure S2. It seemed that PDA is a good choice of wet adhesive which only needs 30 minutes heat treatment for attachment. Please give an explanation of the adhesive mechanism via heat.

Our Response and Revision: We thank the reviewer for noting the selection of polydopamine (PDA) as a “good choice” for strong wet adhesive. PDA was selected because it is known to provide: (1) strong wet adhesion; (2) biocompatibility; (3) durability against harsh mechanical conditions; and (4) conformal coating across uneven surfaces through in-situ polymerization in solution. The bonding mechanism of PDA relies on the irreversible polymerization of dopamine monomer into a thin layer while penetrating the upper surface of commercial soft contact lenses. To elaborate on the bonding mechanism of PDA, we added the following text, “...through the in-situ polymerization of a polydopamine (PDA) adhesive, which is similar to the bonding of marine mussels in nature (Supplementary Fig. 2)²⁵. The in-situ polymerization of the PDA adhesive is irreversible and penetrates into the upper surface of commercial soft contact lenses to form a permanent interaction, which is crucial to ensuring the mechanical durability of the SSCL for long-term routine use.” on page 6; and “After approximately 12 hours, thermal annealing was applied at 60 °C for 30 minutes to complete the in-situ polymerization of the dopamine-hydrochloride into a thin layer of PDA.” on page 22. We also added the following reference.

25. Hu, S. et al. A mussel-inspired film for adhesion to wet buccal tissue and efficient buccal drug delivery. *Nat. Commun.* **12**, 1689 (2021).

Comment #2: Page 8, Line 117-120. I’m wondering the meaning of the capacitive sensitivity. Please explain the meaning and calculation method of “highest capacitive sensitivity ($6.8 \times 10^{-4} \text{ mmHg}^{-1}$)”.

Our Response and Revision: We thank the reviewer for this comment. The capacitive sensitivity (S) presents the capacitive change of the SSCL with respect to the IOP change, which is defined by equation (3). To clarify this aspect, we added the following text, “Details of calculating the capacitive sensitivity are described in the Materials and Methods section.” on page 9; and “The capacitive sensitivity (S) of the SSCL is defined by:

$$S = \frac{\delta((C_2 - C_1)/C_0)}{\delta(P_2 - P_1)} \quad (3)$$

where C_2 and C_1 are the capacitance acquired at sampling events 2 and 1, respectively; C_0 is the initial capacitance; and P_2 and P_1 are the pressure acquired at sampling events 2 and 1, respectively.” on page 23.

Comment #3: Figure 2d & Page 8, Line 125. The label of Figure 2d is incorrectly written as 2c. It seemed that there is an obvious shift after stretching, folding and flipping. Statistical analysis method should be used to clarify the differences. The meaning of error bars need to be defined. The claim “maintained without a substantial shift from baseline” may be not appropriate without statistical significance analysis.

Our Response and Revision: We thank the reviewer for these comments. We corrected the numbering in Figure 2. To define the meaning of error bars (i.e., standard deviation), we added the following text, “A total of 5 measurements were taken and averaged at each data point with the error bars denoting standard deviations.” on page 9; and “The error bars represent standard deviations with n = 5 for each group.” in the caption of Figure 2. We also revised the claim of “maintained without a substantial shift from baseline” into “...changed barely, or slightly within only ± 0.2 MHz, ...” on page 9.

Comment #4: Page 11. P191 & Figure 4b. “the accumulation of proteins on the SSCL remained lower compared to the bare soft contact lens” Why does this happen?

Our Response and Revision: We thank the reviewer for this question. We compared the protein accumulation over the top surface of the internal ocular tonometer of the SSCL as compared to its bare soft contact lens. The internal ocular tonometer is encapsulated with a hydrophobic elastomer (i.e., PDMS) that exhibits a lower affinity to accumulate proteins as compared to the commercial (hydrogel-based) soft contact lenses. To avoid any confusion, we changed the labels in Figures 4b and 4c from “SSCL” to “Ocular Tonometer”. We also revised and added the following text, “...over the top surface of the internal ocular tonometer of the SSCL ...” on page 12; and “The lower accumulation of proteins over the internal ocular tonometer of the SSCL is likely attributed to the presence of its hydrophobic encapsulation layer (i.e., PDMS).” on page 13.

Comment #5: Figure 2f and Figure 2g. Please quantify the results in figure 2f and 2g. While the image shows no significant change in eye response, a quantification of the damage is required.

Our Response and Revision: We thank the reviewer for this comment, which we believe it is referring to Figures 4f and 4g – not Figures 2f and g. To quantify the eye damage, we added Table S1 describing the quantitative assessment of histopathologic inflammation grades of the rabbit cornea and conjunctiva. Accordingly, we also added the following text, “The quantitative assessments of histopathologic inflammation grades in the rabbit cornea and conjunctiva are summarized in Supplementary Table 1.” on page 14.

Group I: 24-Hour Continuous Wear		
Animal ID	Right Eye – SSCL	Left Eye – Bare Lens Control
1	1	2
2	2	1
3	2	2
4	2	1
Group II: 2-Week Wear (8 Hours Per Day)		
	Right Eye – SSCL	Left Eye – Untreated Control
5	0	1
6	1	1

Grades for assessment of inflammation:

0 – None, 1 – Minimal, 2 – Mild, 3 – Moderate, and 4 – Severe.

Supplementary Table 1. Quantitative assessment of histopathologic inflammation grades of the rabbit cornea and conjunctiva.

Comment #6: Page 11, Line 205. The authors describe that after 2 weeks wear (8 hours per day), no or minimal lymphocytic inflammation occurred. And the damage might be background lesions unassociated with the application. In this test, a new clean SSCL was used each day. And the 24 hours wearing resulted grade 1 or 2 lesions. Does that mean the safe duration of SSCL is 8 hours per day? And we know the need for monitoring IOP is more than 8 hours per day.

Our Response and Revision: We thank the reviewer for this comment. In our study, no notable differences were observed in the biosafety (e.g., lesions) and wearability of the SSCL as compared to its bare soft contact lens. The minimal inflammation which appeared in rabbit eyes might be background lesions or caused by the partial suture (i.e., partial temporary tarsorrhaphy) to improve the lens retention at the corneal surface. It is clear that there is not a systematic trend in inflammation association with wearing the SSCL. To clarify this aspect, we revised and added the following text, “The minimal inflammation might be background lesions or caused by the partial suture (i.e., partial temporary tarsorrhaphy) which is unassociated with wearing the SSCL.” on page 14. To further emphasize the biosafety of the SSCL, we also revised and added the following text, “These soft contact lenses can seamlessly fit a variety of corneal shapes and sizes in human eyes without significant safety concerns for those even with glaucoma, other ocular diseases, or post-incisional surgeries²⁴,” on page 5; “..., which have been proven safe for those even with chronic ocular diseases, including glaucoma, or following incisional surgeries²⁴,” on page 8, and “...that have been proven safe for those whether or not the presence of glaucoma or following incisional surgeries.” on page 20. We also added the following reference.

24. Lee, K. Cornea and contact lens considerations in glaucoma. *Modern Optometry* April, 30-32 (2020).

Comment #7: Why choose the different animal model for biocompatibility test (rabbits) and IOP monitoring test (dog)? Are there any special reasons?

Our Response and Revision: We thank the reviewer for this question. We chose both rabbit and dog models specifically to gain insights into the *in vivo* tissue compatibility and measurement accuracy of the SSCL, respectively. In rabbit eyes, we assessed the *in vivo* tissue compatibility and long-term (i.e., weeks) wearability of the SSCL as compared to its bare soft contact lens on the contralateral eye. The rabbits were humanly euthanized for post analysis including hematoxylin and eosin (H&E) staining. In dog eyes, we assessed the *in vivo* 24-hour sensing performance of the SSCL because dogs are easier to handle over rabbits particularly under ambulatory conditions. For calibration, the IOP of the dog eyes was monitored before and after the administration of an IOP-lowering medication (e.g., latanoprost) as compared to control measurements using a conventional rebound tonometer (e.g., I-Care Tonovet; I-Care, Inc.). The dogs were not sacrificed. Both rabbit and dog eyes provide an anatomical similarity to the human eye in shape and size while dogs are much easier to handle over rabbits for IOP monitoring under ambulatory conditions. To clarify this aspect, we revised and added the following text, “The *in vivo* tissue compatibility and long-term wearability of the SSCL were assessed in rabbit eyes (n = 4) ...” on page 13; “The rabbit eye provides an anatomical similarity to the human eye in shape and size²⁷,” on page 13; “The *in vivo* 24-hour sensing performance of the SSCL was assessed in a dog eye (female beagle; 10 months old) under ambulatory conditions...” on page 14; “The dog eye provides an anatomical similarity to the human eye in shape and size²⁷,” on page 14; and “The dog was easier to handle over rabbits for IOP monitoring particularly under ambulatory conditions.” on page 31. We also added the following reference.

27. Bouhenni, R. A., Dunmire, J., Sewell, A. & Edward, D. P. Animal models of glaucoma. *J. Biomed. Biotechnol.* **2012**, 692609 (2012).

Comment #8: Figure 5c. Statistical analysis are supposed to be done for significance of differences between SSCL and iCare, while it seemed there is a large difference at some timepoint, such as 11:00, 9:00.

Our Response and Revision: We thank the reviewer for this comment and allowing us opportunity to clarify this important point. The large differences at some timepoints, such as 11:00 and 9:00, represent abnormal spikes from the measurements with the iCare rather than the SSCL mainly due to motion artifacts even though the dog was forced to remain still with their eyes open. The overall trends of the IOP measures agreed well; within-instrument variability was within only ± 2 mmHg; and approximate maximum deviations across the time period were around 4 mmHg for both instruments. The typically assumed clinically significant difference is around 4 mmHg when considering the inherent noise of IOP measures with any commercial device, even including the GAT. Therefore, we interpreted the overall trends of the

IOP measures to be similar with no significant difference. To clarify this aspect, we revised and added the following text, "...while the iCare Tonovet Plus required the dog to remain still with its eyes open during the measurements. The measurements with the SSCL exhibited a typical IOP rhythm of the dog eye throughout the day in which its overall trends agreed well with those associated with the iCare Tonovet Plus. No notable degradation in signal quality with the SSCL occurred irrespective of blinking or eye movements of the dog. The outliers of the IOP measurements (e.g. from 9 am to 12 pm) using the iCare Tonovet Plus were mainly attributed to motion artifacts by the excitement of the dog." on page 15.

REVIEWER 2

Overall Comments: Excellent paper reporting significant progress in the monitoring of IOP using a passive sensor integrated onto a soft contact lens. The basic idea of a passive RLC sensing device that can be wirelessly interrogated by examining its resonance frequency is similar to what was presented earlier by Ophthimalia (France), but the authors have moved the technology forward so the device is compatible with soft contact lenses. The authors did a thorough job in assessing the device's performance and the basic biocompatibility aspects, also in relation to the competing 'Triggerfish' lens (Sensimed, Switzerland). I only have minor remarks regarding this manuscript.

Our Response: We thank the reviewer for the highly favorable comments and the recommendation for its publication in this journal.

Comment #1: Fig. 2. Benchtop Evaluations: label 'c' is used twice; label 'd' is missing.

Our Response and Revision: We thank the reviewer for this comment. We corrected the numbering of Figure 2.

Comment #2: Line 313: is the improved fit really the major reason for the higher sensitivity? Or is it the difference in rigidity of an *ex vivo* versus an *in vivo* eye as suggested on line 231?

Our Response and Revision: We thank the reviewer for this question. In short, the increased sensitivity is attributed to a combination of the two factors the reviewer has identified. We believe that the improved fit of the SSCL to the eye is the major reason for the higher sensitivity. This tendency was consistent in our measurements in rabbit, dog, and human eyes *in vivo* as compared to those in enucleated pig and cow eyes *ex vivo*. The sensitivity was further decreased in the enucleated pig and cow eyes *ex vivo* due to poor fitting quality of the SSCL across the irregular corneal surface with the insertion of two cannulation needles. To clarify this aspect, we added the following text, "...mainly..." on pages 11 and 20; and "...mainly due to their fitting quality affected by different corneal rigidity and irregularity of the eyes or the *ex vivo* setting with the insertion of cannulation needles³⁰," on page 15

Comment #3: Typo: line 203: 'ribbits' à 'rabbits'

Our Response and Revision: We thank the reviewer for this comment. We corrected the typo.

Comment #4: Methods: (Benchtop evaluations) In the description of how the relative resistance variation (as a result of a number of stretching cycles) and the capacitive response under pressure variations were measured, it is not clear how the AgSEBS layers were electrically contacted.

Our Response and Revision: We thank the reviewer for this comment. For the benchtop measurement of the relative resistance change and capacitive responses, the SSCL was electrically connected using a conducting (i.e., AgSEBS) wire to a source meter (Keithley 2400) and a LCR meter (E4980AL, Keysight), respectively. To clarify this aspect, we added the following text, "For the measurement of the capacitive response, the SSCL was connected to a LCR meter (E4980AL; Keysight, Inc.) via a conducting (i.e., AgSEBS) wire..." on page 23, and "For the measurement of the relative resistance change, the SSCL was connected to a source meter (Keithley 2400; Keysight, Inc.) via a conducting (i.e., AgSEBS) wire..." on page 23.

REVIEWER 3

Overall Comments: Dr. Chi Hwan Lee provided extensive data and results on testing their "smart" contact lens tonometer using an off-the-shelf contact lens. Overall the noteworthy results include: (1) The explanation of the "smart tonometer is clearly explained with encapsulation material, conductive material, adhesives, and reader coil system. (2) Ex vivo studies were conducted to compare IOPs between the "smart contact lens" on whole perfused pig eyes cannulated with a sensor, which is proper method design. It is surprising that the commercial contact lens worked on the pig eyes given the oval shape of their corneas. (3) Pre-clinical data is provided in rabbits, beagle with correlation of IOPs measured with the Tonovet tonometer. While not mentioned, it is assumed the commercial contact lens fit reasonably well for the ex vivo studies, rabbit (i.e., with a tarsorrhaphy) and beagle. The animal models are a standard choice for methods. (4) Proper safety experiments were conducted in vitro and in vivo on the rabbits and in the pilot studies on humans. (5) Durability of the "smart contact lens tonometer" to cleaning, stretching, manipulation of bending to place and remove the contact lens, temperature tolerance, and various contact lens solutions. (6) Use of non-standardized survey to compare comfort of "smart lens" with standard contact lens and Trigger fish. As a reviewer, I believe that this data should be part of the results and not as supplement. (7) Supplementary videos are provided. (8) The supplementary figures are numerous; can any of them be combined? The work presented in this manuscript has the potential to be of great significance for patients with glaucoma to assess IOP variability in the real world beyond limited clinic hours, treatment response to glaucoma medications, treatment response to glaucoma laser, and even in certain glaucoma post-operative settings.

Our Response and Revision: We thank the reviewer for the highly favorable comments and the recommendation for its publication in this journal. As suggested in (6), we included all the supplementary data of the survey results (Figure 6e) in main text. As also suggested in (8), we combined many of the supplementary figures to reduce the number from 17 to 11.

Comment #1: Word choice of "exploded" to describe the "smart lens" composition. Alternative word choices include: "stacked", "layered", "integrated".

Our Response and Revision: We thank the reviewer for this comment. We replaced it with "layered".

Comment #2: A more standard survey to assess comfort would be the use of a Likert scale.

Our Response and Revision: We thank the reviewer for this comment. We believe that there are two important considerations here; the survey question(s) and the way these questions are asked. A Likert scale may provide a similar overall result, but one with decreased sensitivity of the measurement variable, necessitating a larger sample size even in a pilot study to identify any trend. We and other groups have considerable experience with the question(s) asked and the way in which they are asked, so prefer to retain a numeric scale with many possible response choices as opposed to the 5 possible of a Likert scale. This is highly important to clarify this rationale within the text, and therefore we revised and added the following text, "For the quantitative assessment of comfort level, each participant was asked to complete a survey..." in page 17, and "..., in a similar manner as previously used in many clinical studies with wearing contact lenses³³." in page 17. We also added the following reference.

33. Kollbaum, P. S., Jansen, M. E. & Rickert, M. E. Comparison of patient-reported visual outcome methods to quantify the perceptual effects of defocus. *Cont. Lens Anterior Eye* **35**, 213-221 (2012).

Comment #3: For supplementary Fig 15, it is not clear what the two dashed lines are for; are these confidence intervals?

Our Response and Revision: We thank the reviewer for this question. The dashed lines denote the range of IOP under each postural condition in prior studies. The results show that our IOP measures agreed well

with those of prior studies. To clarify this point, we added the following text, “The dashed lines denote the range of IOP under each postural condition in prior studies.” in the figure caption.

REVIEWERS' COMMENTS

Reviewer #1 (Remarks to the Author):

Thanks the authors for the responses and revision. I don't have questions any more and recommend the publication of this work in Nature Communications.

Reviewer #2 (Remarks to the Author):

The authors have adequately responded to my comments and revised the manuscript in a satisfactory manner.

Therefore I recommend publication of the manuscript in its present state.

Reviewer #3 (Remarks to the Author):

The authors provided point-by-point clarifications and revisions to each of the three reviewers comments and questions.

As noted in the first reviews, these authors have made significant progress in use of contact lens-based technology for IOP sensing that is not dependent upon the limited office-based tonometry. The proper steps to provided data on sensitivity and specificity of IOP measurements and safety in preclinical animal models of rabbit and beagle are provided. Similar preliminary data in humans are also provided. These data are the essential steps before a clinical trial.

The bigger question that remains to be determined in the future is the relative ease with which such technology will be adopted for clinical use.

RESPONSE TO REVIEWER 1

Reviewer's Comment:

Thanks the authors for the responses and revision. I don't have questions any more and recommend the publication of this work in Nature Communications.

Our Response:

We thank the reviewer for the recommendation of publishing our manuscript in this journal.

RESPONSE TO REVIEWER 2

Reviewer's Comment:

The authors have adequately responded to my comments and revised the manuscript in a satisfactory manner. Therefore I recommend publication of the manuscript in its present state.

Our Response:

We thank the reviewer for the recommendation of publishing our manuscript in this journal.

RESPONSE TO REVIEWER 3

Reviewer's Comment:

The authors provided point-by-point clarifications and revisions to each of the three reviewers comments and questions.

As noted in the first reviews, these authors have made significant progress in use of contact lens-based technology for IOP sensing that is not dependent upon the limited office-based tonometry. The proper steps to provided data on sensitivity and specificity of IOP measurements and safety in preclinical animal models of rabbit and beagle are provided. Similar preliminary data in humans are also provided. These data are the essential steps before a clinical trial. The bigger question that remains to be determined in the future is the relative ease with which such technology will be adopted for clinical use.

Our Response:

We thank the reviewer for the highly favorable comments and the recommendation of publishing our manuscript in this journal. As the reviewer commented, we will move forward this technology into a large scale of clinical study to be adopted for future clinical use and beneficial to millions of glaucoma patients in need.